# Can Vision–Language Models Assess Graphic Design Aesthetics? A Benchmark, Evaluation, and Dataset Perspective

**Arctanx An**[1,2*]  **Shizhao Sun**[2†]  **Danqing Huang**[3]  **Mingxi Cheng**[3]
**Yan Gao**[4]  **Ji Li**[3]  **Yu Qiao**[4]  **Jiang Bian**[2]
[1]Peking University  [2]Microsoft Research Asia  [3]Microsoft  [4]Central South University

## Abstract

Assessing the aesthetic quality of graphic design is central to visual communication, yet remains underexplored in vision–language models (VLMs). We investigate whether VLMs can evaluate design aesthetics in ways comparable to humans. Prior work faces three key limitations: benchmarks restricted to narrow principles and coarse evaluation protocols, a lack of systematic VLM comparisons, and limited training data for model improvement. In this work, we introduce AesEval-Bench, a comprehensive benchmark spanning four dimensions, twelve indicators, and three fully quantifiable tasks: aesthetic judgment, region selection, and precise localization. Then, we systematically evaluate proprietary, open-source, and reasoning-augmented VLMs, revealing clear performance gaps against the nuanced demands of aesthetic assessment. Moreover, we construct a training dataset to fine-tune VLMs for this domain, leveraging human-guided VLM labeling to produce task labels at scale and indicator-grounded reasoning to tie abstract indicators to concrete design regions. Together, our work establishes the first systematic framework for aesthetic quality assessment in graphic design. Our code and dataset will be released at: https://github.com/arctanxarc/AesEval-Bench.

## 1 Introduction

The rapid development of *vision-language models (VLMs)* (Yang et al., 2025; Li et al., 2024a; Hurst et al., 2024) has opened new opportunities for understanding and acting upon multimodal information. While they have already achieved remarkable progress in traditional vision tasks such as image captioning (Lin et al., 2024b; Li et al., 2025b) and visual question answering (An et al., 2024; Zhang et al., 2024; Lin et al., 2025), VLMs are increasingly expected to contribute to new applications. Among these, *graphic design*—which integrates textual and visual elements to convey information across advertising, branding, and digital media—represents a promising direction given its broad societal and practical impact. Crucially, the success of graphic design critically depends on its *aesthetic quality*, shaped by principles such as balance, contrast, and hierarchy.

In this work, we aim to explore a central question: *can VLMs understand and evaluate aesthetic quality of graphic design in a manner comparable to humans?* This question is of significant importance for at least three reasons. First, it can assist human designers by identifying where a design falls short and explaining why, thereby enabling more effective improvement. Second, for generative AI systems, it provides the basis for automatic feedback loops that can guide iterative refinement without extensive human intervention. Third, it suggests an opportunity to extend VLMs beyond factual recognition toward aesthetic evaluation.

Despite its importance, research along this direction remains limited (see Table 1). First, *benchmarks are inadequate*. Those (Zhou et al., 2024; Huang et al., 2024) developed for natural photos often ignore design-specific factors such as typography, while early ones for graphic design typically cover only a narrow subset of design principles (Lin et al., 2023; 2024a; Jiang & Chen, 2025). Moreover,

---

*Work was done during an internship at Microsoft Research Asia
†Project Leader

existing evaluation protocols are limited. Scoring-based methods fail to indicate where poor aesthetics occur (Haraguchi et al., 2024), while description-based ones provide qualitative feedback that is difficult to quantify (Jung et al., 2025). Second, *comparisons between VLMs are missing*. There has been no systematic evaluation across different VLMs, whether open-source or closed-source, in the context of design aesthetics. Third, *training datasets are lacking*, leaving the question of how to further improve VLM performance in this domain underexplored.

As a first step toward addressing these limitations, we introduce *AesEval-Bench*, a new benchmark for evaluating the aesthetic quality of graphic designs (see Figure 1). Drawing on prior literature (Li & Chen, 2009; Wangwiwattana & Meeyen, 2024), we identify four critical dimensions—typography, layout, color, and graphics—that together comprehensively capture design aesthetics. These dimensions reflect the major factors that humans consistently emphasize when assessing visual appeal. To provide finer granularity, we further define twelve indicators that specify concrete aspects within each dimension, such as hierarchy and legibility under typography. For each indicators, we then design three challenging tasks: 1) *aesthetic judgment*, which asks models to decide whether a design is aesthetically pleasing (yes/no), providing a straightforward measure of overall perception; 2) *region selection*, which requires models to choose from candidate regions where unpleasing elements appear, testing their ability to pinpoint problematic areas beyond a global judgment; 3) *precise localization*, which challenges models to predict the exact bounding box (bbox) coordinates of unpleasing areas, offering the most detailed diagnosis and reflecting a deeper understanding of aesthetics. Unlike prior benchmarks (), AesEval-Bench not only covers a broad range of aesthetic factors through its dimensions and indicators, but also defines evaluation tasks that are fully quantifiable via choice or bbox prediction formats, enabling systematic and reproducible assessment of design aesthetics.

With AesEval-Bench, we systematically evaluate VLMs on their ability to assess the aesthetic quality of graphic designs (see Table 1). For each design, models perform three tasks across twelve indicators. We use human annotation as ground truth, and measure performance by accuracy (for aesthetic judgment and region selection) and bbox IoU (for precise localization). Overall, our results reveal clear gaps between current state-of-the-art VLMs and the nuanced demands of aesthetic quality assessment. Specifically, proprietary VLMs (e.g., GPT series (Achiam et al., 2023)) outperform open-source ones (e.g., Qwen-VL (Wang et al., 2024), Intern-VL (Zhu et al., 2025), LLaVA (Liu et al., 2023)). Among open-source models, larger variants (32B, 72B) generally achieve better performance than smaller ones (7B). Surprisingly, reasoning-augmented VLMs (e.g., GPT-o1 (Jaech et al., 2024), GPT-o3, Gemini-2.5-Pro (Comanici et al., 2025)) offer no clear advantage over their non-reasoning counterparts. Together, these findings expose the limitations of existing VLMs and underscore the need for domain-specific training tailored to aesthetic quality assessment.

Building on these findings, we move beyond evaluation and turn to training VLMs for aesthetic quality assessment. To this end, we construct a training dataset consisting of three components: the *task* (what the model is asked to do), the *task label* (the expected answer), and the *reasoning path* (the explanation leading to the answer) (see Figure 2). We treat the reasoning path as essential, since generic reasoning has shown little benefit. However, constructing such data poses two key challenges: producing task labels at scale is costly, and generating reasoning paths that genuinely improve performance requires new approaches. To address these, we introduce two solutions. First, *human-guided VLM labeling*, where a small set of human annotations serve as in-context examples to instruct powerful VLMs in producing task labels. This approach maintains alignment with human understanding while reducing manual annotation costs. Second, *indicator-grounded reasoning*, where abstract indicators (e.g., hierarchy or layering) are explicitly tied to concrete regions in the design. Each reasoning path consists of bounding-box coordinates linked to the indicator and textual explanations for their relevance, providing fine-grained and interpretable supervision. We fine-tune VLMs with the task as input and both the reasoning path and the task label as supervision, and evaluate on AesEval-Bench. The results show consistent performance gains across all tasks and indicators (e.g., 5.97%, 2.70% and 17.17%), demonstrating that human-guided VLM labeling yields reliable labels and indicator-grounded reasoning supplies effective supervision.

To sum up, our contributions are as follows:

- We introduce AesEval-Bench, a comprehensive benchmark for assessing aesthetic quality of graphic designs spanning four dimensions, twelve indicators and three quantifiable tasks.
- We systematically evaluate proprietary, open-source, and reasoning-augmented VLMs, revealing clear performance gaps in aesthetic quality assessment.

Table 1: A comparison of AesEval-Bench with existing benchmarks for both image aesthetics and design aesthetics. We highlight key differences in their scale, task formats, source data, covered design dimensions, and the inclusion of reasoning paths.

| Benchmark | #Data | Task Format | Source | Source Type | Font | Layout | Graphics | Color | Training Set | Reasoning Path | Open-source |
|---|---|---|---|---|---|---|---|---|---|---|---|
| | | | | | \multicolumn{4}{c} Dimension | | | | | | |
| *Image Aesthetics Benchmark* | | | | | | | | | | | |
| AesBench (Huang et al., 2024) | ~10k | Free-form | Photographic Image | Image-only | × | × | ✓ | ✓ | × | × | ✓ |
| UNIAA-Bench (Zhou et al., 2024) | ~6k | Free-form | Photographic Image | Image-only | × | ✓ | × | ✓ | × | × | ✓ |
| FineArtBench (Jiang & Chen, 2025) | - | Choice Free-form | Art Work + Photographic Image | Image-only | × | × | × | ✓ | × | ✓ | × |
| *Design Aesthetics Benchmark* | | | | | | | | | | | |
| DesignBench (Lin et al., 2023) | - | Choice Free-form | Graphic Design | Image+Json | ✓ | ✓ | × | ✓ | × | × | ✓ |
| DesignProbe (Lin et al., 2024a) | ~1.6k | Choice | Graphic Design | Image+Json | ✓ | ✓ | × | ✓ | × | × | × |
| GPT-Eval Bench (Haraguchi et al., 2024) | ~2k | Scoring | Graphic Design | Image+Json | × | ✓ | × | × | × | × | × |
| UI-Bench (Jung et al., 2025) | ~3k | Choice Description | UI Design | Image-only | ✓ | ✓ | × | ✓ | × | × | × |
| UICrit (Jung et al., 2025) | ~3k | Free-form Bbox regression | UI Design | Image-only | ✓ | ✓ | × | ✓ | × | × | ✓ |
| AesEval-Bench (Ours) | ~4.5k | Choice Bbox Regression | Graphic Design | Image+Json | ✓ | ✓ | ✓ | ✓ | ✓ | ✓ | ✓ |

- We construct a training dataset to fine-tune VLMs for this domain. Our approach introduces human-guided VLM labeling to produce task labels at scale and indicator-grounded reasoning to tie abstract indicators to concrete design regions. Experiments on AesEval-Bench show that this dataset consistently improves performance across all tasks.

## 2 RELATED WORKS

**Aesthetic Quality Assessment.** Aesthetic quality assessment (Deng et al., 2017) aims to automatically evaluate visual appeal, serving as a computational proxy for human judgment. Within this area, two major lines of research have emerged (see Table 1). *Image aesthetics assessment* (Huang et al., 2024; Zhou et al., 2024; Jiang & Chen, 2025) focuses on photographic images, where quality is determined by factors such as color harmony, lighting, and subject placement. *Design aesthetics assessment* (Lin et al., 2023; 2024a; Haraguchi et al., 2024; Jung et al., 2025) targets graphic designs such as posters, advertisements, or user interfaces, which depend on design-related factors including typography, hierarchy, and alignment. Our work falls within design aesthetics assessment.

Despite its importance, design aesthetics assessment remains underexplored. Existing benchmarks capture only a narrow subset of design dimensions. For instance, (Lin et al., 2024a) omits graphics-related factors, while (Haraguchi et al., 2024) ignores both fonts and graphics. Furthermore, their task formulations lack rigor. Some adopt free-form question answering, which is difficult to quantify (Lin et al., 2023), while others provide only holistic scores without identifying problematic regions, limiting interpretability and actionability (Jung et al., 2025). Our work introduces a benchmark that comprehensively covers design-related aesthetic factors across font, layout, graphics and color, defining well-structured and quantifiable tasks using choice and bbox prediction formats.

**Vision-Language Models.** Vision-Language Models (VLMs) (Wang et al., 2024; Comanici et al., 2025; Li et al., 2024a; Zhang et al., 2025c) have achieved remarkable performance on tasks such as image captioning (Luo et al., 2025a; Li et al., 2024b; Zhang et al., 2025d) and visual question answering (An et al., 2024; Lin et al., 2024b; Luo et al., 2025b). Yet, their ability to assess the aesthetic quality of graphic designs remains largely unexplored. Prior work typically evaluates only one or two VLMs (e.g., (Haraguchi et al., 2024) studies GPT). We provide a systematic comparison across a broad set of VLMs, including proprietary, open-source, and reasoning-augmented models.

Recently, increasing attention has been devoted to the reasoning capabilities of VLMs (Li et al., 2025a; Zhang et al., 2025b). For example, (Sarch et al., 2025) employs tree-based search to improve reasoning chains, (Li et al., 2025a) visualizes reasoning trajectories for transparency, and (Sun et al., 2024; Shao et al., 2024; Cao et al., 2025; Wu et al., 2025) explores grounded visual reasoning by jointly generating bounding boxes and textual explanations. In our work, we observe that generic reasoning in current VLMs provides limited benefit for assessing design aesthetics. To address this, we construct a training dataset with reasoning paths that explicitly link abstract design indicators

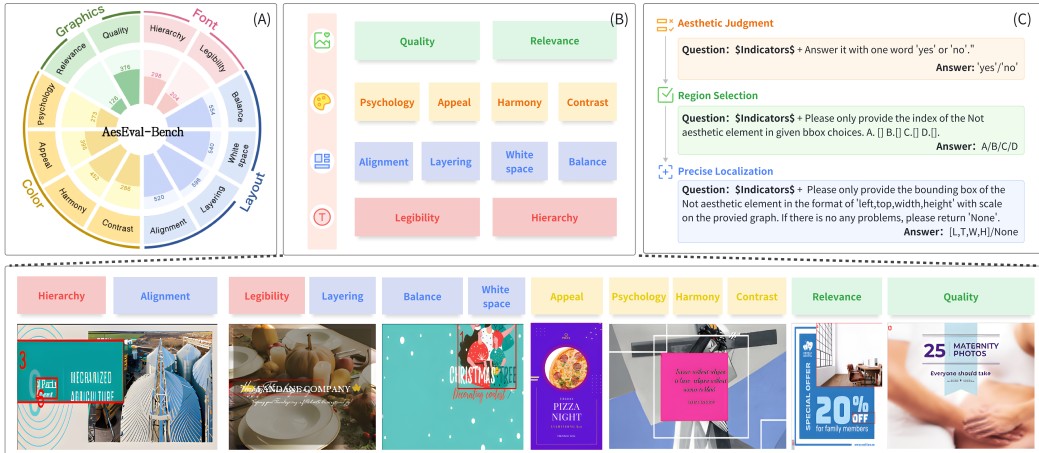

Figure 1: Overview of AesEval-Bench. **(A)** The four dimensions and twelve indicators considered in the benchmark. Numbers inside the circles indicate how many designs are labeled as flawed for each indicator. **(B)** Example designs illustrating the indicators, with regions exhibiting aesthetic issues highlighted by red boxes. Detailed textual explanations of all indicators are provided in the Appendix. **(C)** The three tasks, along with example questions and their expected answers.

to concrete regions of the design. Unlike grounded visual reasoning, which localizes semantically salient entities (e.g., a "dog" or "chair"), our regions are indicator-centric, capturing higher-level concepts such as hierarchy, alignment, and spacing that directly embody design principles.

## 3 BENCHMARK CONSTRUCTION

### 3.1 OVERVIEW

**AesEval-Bench** formulates design aesthetics assessment as a question–answering task. The input contains a *task* description and a *design image*, optionally accompanied by metadata such as layout, font, or color information in JSON format. The output is the *answer* corresponding to the task.

To capture different aspects of design aesthetics assessment, we introduce three task types (Figure 1(C)): 1) *aesthetic judgment* asks whether a design is aesthetically pleasing (yes/no), providing a measure of overall perception. 2) *region selection* requires choosing from candidate regions where aesthetic issues appear, testing the ability to localize problematic areas beyond a global judgment. 3) *precise localization* requires predicting the exact bounding box coordinates of problematic regions, offering a fine-grained diagnosis. Each task is accompanied by the explanation of an indicator—the key factor humans consistently emphasize when evaluating visual appeal (e.g., hierarchy, layering, contrast). We consider twelve indicators (Figure 1(B)), grouped into four dimensions (Figure 1(A)).

For design images, we sample 1200 designs from the test split of Crello dataset (Yamaguchi, 2021), which contains professional graphic designs with both the design image and its metadata. The expected answers differ across tasks. For aesthetic judgment, the answer is yes or no. For region selection, it is the index of one region among four candidates. For precise localization, it is the bounding box coordinates of the identified region or None if the design has no aesthetic issues.

Overall, AesEval-Bench comprises 4500 base question–answer pairs (three tasks across 1500 designs), each further instantiated across twelve indicators to enable fine-grained evaluation.

### 3.2 CURATION PIPELINE

**Establishing Dimensions and Indicators.** Aesthetic quality in graphic design is inherently multidimensional. To define a rigorous benchmark, we first conducted a comprehensive literature review of classical and contemporary design principles (McCormack & Lomas, 2020; Lou et al., 2022; Lu et al., 2020). We then consulted professional designers to refine this taxonomy, ensur-

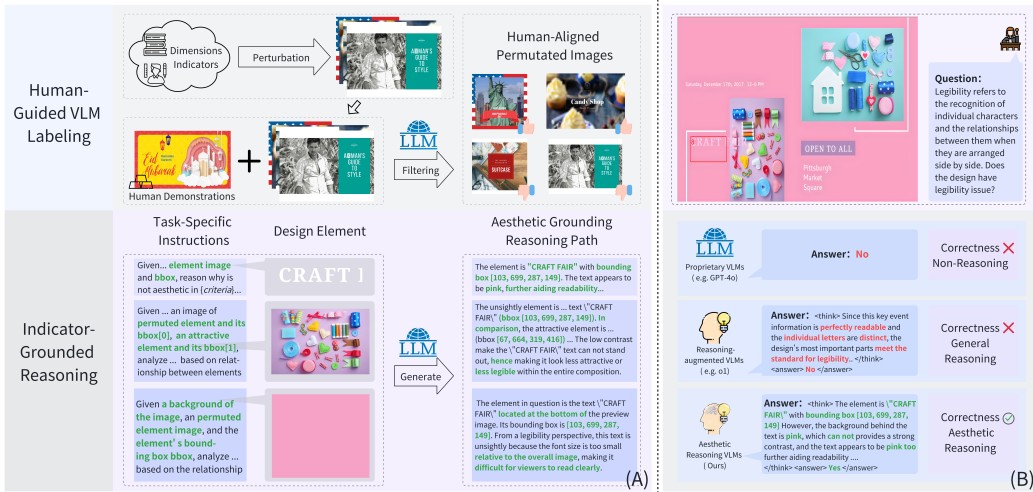

Figure 2: **(A)** Illustration of two key steps in training data construction. Human-guided VLM labeling enables scalable determination of whether designs exhibit aesthetic issues. Indicator-grounded reasoning generates reasoning paths that explicitly link abstract indicators to concrete design regions (represented as bbox coordinates). **(B)** Example highlighting the difference between non-reasoning models, generic reasoning models, and our indicator-grounded reasoning model.

ing alignment with both theoretical foundations and practical expertise. This process yielded four core dimensions—layout, font, graphics, and color (Figure 1(A))—each further specified by concrete indicators consistently emphasized in human aesthetic judgment. In total, we distilled twelve indicators that together capture the essential factors of design aesthetics (Figure 1(B)).

**Constructing Potentially Flawed Designs.** As introduced in Section 3.1, the design images in AesEval-Bench are sourced from the Crello dataset, which contains professional-quality graphic designs. To effectively evaluate design aesthetics, the benchmark must include not only well-designed but also less appealing examples. We therefore repurpose Crello by introducing controlled perturbations, such as repositioning elements, altering font choices, or adjusting colors. These perturbations may either degrade the visual quality or leave it largely intact. For example, slightly enlarging a heading might preserve hierarchy, whereas shifting it left could disrupt balance. Each base design undergoes one to three random perturbations, generating a spectrum of variations that range from aesthetically unchanged to noticeably flawed, while still appearing realistic. Since Crello Yamaguchi (2021) provides element-level metadata in JSON format along with separate design layers, these perturbations can be applied directly at the JSON level and rendered into new design images by recombining the modified metadata with the corresponding layers.

**Human-in-the-Loop Aesthetic Review.** We engage human annotators to verify whether the perturbed designs truly exhibit aesthetic issues. Before annotation, all annotators receive a tutorial that includes examples of both well-designed and flawed cases, along with detailed explanations of the underlying reasons. During the review, each annotator is shown a design image together with a description of the focal indicator and asked to determine whether the design contains the corresponding flaw (yes or no). For each design, we derive the final label by applying majority voting across multiple annotators to ensure consensus.

**Generating Question-Answer Pairs.** With metadata in JSON format, records of applied perturbations and human annotations, we can systematically construct answers corresponding to each task. For aesthetic judgment, the rule is straightforward: if human annotators label a design as good, the ground-truth answer is no (i.e., no aesthetic issue); otherwise, it is yes. For region selection, if a design is labeled as good, the four answer choices consist of three randomly sampled bboxes from the metadata and a None option, with the ground-truth answer being None. If the design is labeled as flawed, the four choices include the bbox of the perturbed element, two randomly sampled bboxes from the metadata, and None, with the ground-truth answer set to the bbox of the perturbed element. For precise localization, if a design is labeled as good, the ground-truth answer is None; otherwise, it corresponds to the exact bbox of the perturbed element.

### 3.3 EVALUATION PROTOCOLS

For aesthetic judgment and region selection, both formulated as choice problems, we adopt accuracy as the metric, measuring the exact match between model predictions and the ground truth. For precise localization, the task combines two components: a choice problem (predicting None when no aesthetic issue exists) and a bounding box regression problem (predicting the exact bbox when an issue is present). Accordingly, we use accuracy for cases where the ground truth is None, and intersection over union (IoU)—which quantifies the overlap between the predicted and ground-truth bboxes—for cases where a bbox is required.

## 4 TRAINING DATA CONSTRUCTION

Evaluation on popular VLMs reveals clear gap between the capabiliteis of current state-of-the-arts VLMs and the nuanced requirements of aesthetic quality assessment. Moreover, reasoning-augmented VLMs show no clear performance gains (see Section 5.1).

To this end, we construct a training dataset, named **AesEval-Train**, to fine-tune VLMs for this domain. First, we adopt the same procedure as benchmark construction to *construct potentially flawed designs* (see Section 3.2). Next, since relying solely on human annotation to determine whether perturbed designs exhibit aesthetic issues is neither scalable nor cost-effective for training at large scale, we introduce *human-guided VLM labeling*. Then, we follow the benchmark construction to *generate question–answer pairs* (see Section 3.2). Finally, we introduce *indicator-grounded reasoning* to generate domain-specific reasoning paths aimed at improving task performance. In the following, we describe in detail the two steps that differ from benchmark construction.

**Human-Guided VLM Labeling.** We leverage a small set of human annotations as demonstrations, together with the bbox coordinates of perturbed regions, as input to strong VLMs. The model is instructed to generate a binary label indicating whether the perturbed design exhibits an aesthetic issue (see Figure 2(A)). By incorporating human annotations, we preserve alignment with human judgment while substantially reducing manual annotation costs. Moreover, providing the perturbation region as a prior, which is unavailable in real-world scenarios, simplifies the labeling process and improves reliability. With these two sources of guidance, while the generated labels may not be perfectly accurate, they yield a training set of sufficient quality to enhance fine-tuning performance.

**Indicator-Grounded Reasoning.** As illustrated in Figure 2(B), generic reasoning often explains or analyzes a given indicator and task without grounding the discussion in relevant regions of the design. To address this limitation, we propose explicitly linking abstract indicators to concrete regions within the design. Specifically, we include both the bounding box (bbox) coordinates of relevant regions and textual explanations of their relevance to the indicator in the reasoning path.

To obtain such reasoning paths, we instruct powerful VLMs (e.g., GPT in our experiments) by providing them with the bbox coordinates of the target regions and the corresponding design layers (Figure 2(A)). The model is required to output the provided coordinates alongside an explanation of how the region relates to the indicator, thereby ensuring that the reasoning path consistently contains the desired information. We further adopt task-specific strategies to determine the regions of interest. For aesthetic judgment, we directly use the bbox of the perturbed regions. For region selection, we include both the perturbed and non-perturbed regions to strengthen the model's ability to discriminate among candidate regions. For precise localization, we not only highlight the bbox of perturbed regions but also emphasize their relationship to the overall design, enabling the model to better localize problematic regions within the global design context.

## 5 EXPERIMENT

### 5.1 BENCHMARKING VLMS ON AESEVAL-BENCH

**Setups.** We conduct a comprehensive evaluation of 10 VLMs spanning diverse model families and parameter scales. For non-reasoning models, we consider open-source representatives such as LLaVA (Liu et al., 2023), Qwen2.5-VL (Bai et al., 2025), and Intern-VL3 (Zhu et al., 2025), as well as closed-source GPT models (Jaech et al., 2024). For reasoning-augmented models, we evaluate

Table 2: Evaluation on aesthetic judgment task. Overall acc is the average value of all indicators. The best and second-best results are highlighted in **bold** and underlined, respectively.

| Model | Overall Acc | Layout | | | | Color | | | | Font | | Graphics | |
|---|---|---|---|---|---|---|---|---|---|---|---|---|---|
| | | balance | layering | whitespace | alignment | harmony | contrast | appeal | psycholoy | legibility | hierarchy | quality | relevance |
| **Non-reasoning Models** | | | | | | | | | | | | | |
| LLaVA-13B | 0.5636 | 0.6506 | 0.5063 | 0.6975 | 0.6759 | 0.4411 | 0.2409 | 0.2615 | 0.2851 | 0.7660 | 0.8260 | 0.7386 | 0.6733 |
| Qwen-VL-7B | 0.6390 | 0.8272 | 0.4508 | 0.8076 | 0.8413 | 0.8223 | 0.9136 | 0.4009 | 0.1355 | 0.9430 | 0.3100 | 0.8183 | 0.3970 |
| Qwen-VL-32B | 0.6458 | 0.6762 | 0.5689 | 0.6862 | 0.6941 | 0.6209 | 0.5776 | 0.4948 | 0.1862 | 0.8023 | 0.7618 | 0.7708 | 0.9001 |
| Qwen-VL-72B | 0.6724 | 0.6752 | 0.6925 | 0.7131 | 0.6731 | 0.6921 | 0.7804 | 0.2620 | 0.2447 | 0.8739 | 0.7952 | 0.7729 | 0.8734 |
| Intern-VL3-8B | 0.6331 | 0.4617 | 0.6452 | 0.7577 | 0.7751 | 0.3487 | 0.3329 | 0.3082 | 0.5180 | 0.9486 | 0.6951 | 0.8571 | 0.9491 |
| Intern-VL3-14B | 0.6883 | 0.7406 | 0.3826 | 0.7563 | 0.7601 | 0.7706 | 0.8594 | 0.5276 | 0.1912 | 0.8309 | 0.8304 | 0.8373 | 0.7729 |
| GPT-4o | 0.7031 | 0.7588 | 0.6789 | 0.6597 | 0.4688 | 0.8129 | 0.8190 | 0.8237 | 0.3292 | 0.7506 | 0.7344 | 0.7857 | 0.8160 |
| GPT-5 | **0.7252** | 0.8378 | 0.7832 | 0.7275 | 0.6510 | 0.6953 | 0.8375 | 0.4000 | 0.5237 | 0.7472 | 0.7419 | 0.9023 | 0.8551 |
| **Reasoning-augmented Models** | | | | | | | | | | | | | |
| GPT-o1 | 0.6705 | 0.7384 | 0.7531 | 0.5522 | 0.3398 | 0.7149 | 0.5439 | 0.7427 | 0.6750 | 0.7049 | 0.7266 | 0.7518 | 0.8030 |
| GPT-o3 | 0.7105 | 0.7450 | 0.7597 | 0.6588 | 0.3964 | 0.7715 | 0.7005 | 0.7993 | 0.6316 | 0.7615 | 0.7332 | 0.8084 | 0.7596 |
| Gemini-2.5-Pro | 0.6368 | 0.7355 | 0.6924 | 0.5936 | 0.5089 | 0.7333 | 0.6495 | 0.6776 | 0.5604 | 0.5888 | 0.6997 | 0.5217 | 0.6803 |
| **Expert Models for Image Aesthetics Assessment** | | | | | | | | | | | | | |
| AesExpert-7B | 0.4056 | 0.5025 | 0.4142 | 0.3317 | 0.4636 | 0.3017 | 0.4147 | 0.3253 | 0.2318 | 0.2670 | 0.5073 | 0.6166 | 0.4904 |
| UNIAA-LLaVA | 0.2900 | 0.2393 | 0.2120 | 0.2471 | 0.2207 | 0.2733 | 0.3316 | 0.3041 | 0.3073 | 0.2893 | 0.5266 | 0.2410 | 0.2879 |

Table 3: Evaluation on region selection task. Overall acc is the average value of all indicators. The best and second-best results are highlighted in **bold** and underlined, respectively.

| Model | Overall Acc | Layout | | | | Color | | | | Font | | Graphics | |
|---|---|---|---|---|---|---|---|---|---|---|---|---|---|
| | | balance | layering | whitespace | alignment | harmony | contrast | appeal | psycholoy | legibility | hierarchy | quality | relevance |
| **Non-reasoning Models** | | | | | | | | | | | | | |
| LLaVA-13B | 0.6065 | 0.5823 | 0.5713 | 0.5612 | 0.5918 | 0.6171 | 0.5166 | 0.6319 | 0.6856 | 0.6130 | 0.7003 | 0.6329 | 0.5745 |
| Qwen-VL-7B (Base) | 0.5795 | 0.5128 | 0.5370 | 0.5748 | 0.5443 | 0.5822 | 0.5433 | 0.5412 | 0.6384 | 0.5974 | 0.6379 | 0.6258 | 0.6190 |
| Qwen-VL-32B | 0.6311 | 0.5933 | 0.5397 | 0.5012 | 0.5252 | 0.5678 | 0.7065 | 0.6367 | 0.5735 | 0.7833 | 0.6278 | 0.7650 | 0.7533 |
| Qwen-VL-72B | 0.6626 | 0.5105 | 0.5839 | 0.4547 | 0.5348 | 0.5977 | 0.7225 | 0.6495 | 0.7940 | 0.7728 | 0.7934 | 0.7360 | 0.8015 |
| Intern-VL3-8B | 0.5799 | 0.5242 | 0.4948 | 0.5606 | 0.5363 | 0.5527 | 0.5568 | 0.5805 | 0.6342 | 0.6583 | 0.6157 | 0.6031 | 0.6415 |
| Intern-VL3-14B | 0.6378 | 0.5872 | 0.5204 | 0.5945 | 0.5745 | 0.6282 | 0.6997 | 0.6419 | 0.7034 | 0.7244 | 0.6870 | 0.6109 | 0.6818 |
| GPT-4o | 0.6745 | 0.4714 | 0.4894 | 0.5007 | 0.6011 | 0.7406 | 0.8591 | 0.7166 | 0.8135 | 0.6633 | 0.9080 | 0.6444 | 0.6865 |
| GPT-5 | **0.6989** | 0.6484 | 0.5929 | 0.6630 | 0.6396 | 0.7229 | 0.7214 | 0.7510 | 0.6847 | 0.8038 | 0.7565 | 0.6953 | 0.7070 |
| **Reasoning-augmented Models** | | | | | | | | | | | | | |
| GPT-o1 | 0.6347 | 0.6319 | 0.5746 | 0.6397 | 0.5934 | 0.6178 | 0.6323 | 0.6880 | 0.7734 | 0.6320 | 0.7092 | 0.5936 | 0.5305 |
| GPT-o3 | 0.6581 | 0.6483 | 0.6263 | 0.5325 | 0.3653 | 0.8272 | 0.5486 | 0.7113 | 0.7981 | 0.6586 | 0.7744 | 0.6601 | 0.7466 |
| Gemini-2.5-Pro | 0.6100 | 0.6810 | 0.6981 | 0.6096 | 0.2992 | 0.6050 | 0.6538 | 0.6678 | 0.6050 | 0.6164 | 0.6539 | 0.5696 | 0.6605 |
| **Expert Models for Image Aesthetics Assessment** | | | | | | | | | | | | | |
| AesExpert-7b | 0.2883 | 0.2954 | 0.2280 | 0.3174 | 0.2631 | 0.3426 | 0.2646 | 0.3176 | 0.3176 | 0.3176 | 0.2588 | 0.2765 | 0.2602 |
| UNIAA-LLaVA | 0.2418 | 0.1619 | 0.2915 | 0.1552 | 0.4075 | 0.1700 | 0.1516 | 0.1760 | 0.2479 | 0.2777 | 0.3796 | 0.2861 | 0.1968 |

GPT-o1, GPT-o3, and Gemini-2.5-Pro (Comanici et al., 2025). In addition, we include expert models specifically designed for image aesthetic assessment, namely AesExpert (Huang et al., 2024) and UNIAA-LLAVA (Zhou et al., 2024). All models are evaluated under the same input setting, which consists of a question (see Figure 1), a design image, and metadata in JSON format.

**Results.** The performance of VLMs are evaluated following the protocols introduced in Section 3.3. Specifically, in addition to reporting scores for each individual indicator, we also provide an overall score computed as the average across all indicators.

*Aesthetic Judgment.* Table 2 presents the results. First, among non-reasoning models, GPT-5 achieves the highest performance, with an overall accuracy of 0.7252. This suggests that even the strongest VLMs still struggle with design aesthetics assessment. Second, reasoning-augmented models do not outperform their non-reasoning counterparts (e.g., GPT-o1 Jaech et al. (2024) and GPT-o3 vs. GPT-4o Hurst et al. (2024) and GPT-5), indicating that generic reasoning provides little benefit in this domain. Third, expert models designed for image aesthetics assessment perform worse overall, highlighting a substantial gap between design aesthetics and image aesthetics. Finally, model performance varies across indicators. For instance, the Qwen-VL series tends to perform better on legibility but worse on psychology compared to other VLMs.

*Region Selection.* Table 3 reports the results. First, VLM performance on this task is generally worse than on aesthetic judgment, likely because it requires not only assessing whether a design is pleasing but also identifying where flaws occur. Second, consistent with aesthetic judgment, GPT-5 achieves the best performance, while reasoning-augmented models show no clear advantage. Finally, across model families, larger models (e.g., 72B) typically outperform smaller ones (e.g., 7B). We find some model has the phenomenon of overfitting, so we adopt a weighted sum when calculating final score.

Table 4: Evaluation on precise localization task for the choice component where the model should predict None if no aesthetic issues are present. Overall score is the average accuracy of all indicators. The best and second-best results are highlighted in **bold** and underlined, respectively.

| Model | Overall Score | Layout | | | | Color | | | | Font | | Graphics | |
|---|---|---|---|---|---|---|---|---|---|---|---|---|---|
| | | balance | layering | whitespace | alignment | harmony | contrast | appeal | psycholoy | legibility | hierarchy | quality | relevance |
| **Non-reasoning Models** | | | | | | | | | | | | | |
| LLaVA-13B | 0.4455 | 0.4523 | 0.6130 | 0.5699 | 0.4714 | 0.3356 | 0.3723 | 0.2898 | 0.2474 | 0.7301 | 0.2453 | 0.5611 | 0.4573 |
| Qwen-VL-7B | 0.5192 | 0.5104 | 0.5839 | 0.4546 | 0.5347 | 0.4376 | 0.5625 | 0.5495 | 0.5339 | 0.4528 | 0.5334 | 0.5360 | 0.5415 |
| GPT-4o | 0.5680 | 0.6417 | 0.2063 | 0.5954 | 0.1679 | 0.8626 | 0.6372 | 0.7200 | 0.6713 | 0.4594 | 0.5164 | 0.7762 | 0.5618 |
| GPT-5 | **0.6090** | 0.6306 | 0.6142 | 0.6910 | 0.6247 | 0.6057 | 0.6170 | 0.5643 | 0.5989 | 0.5464 | 0.6134 | 0.6217 | 0.5804 |
| **Reasoning-augmented Models** | | | | | | | | | | | | | |
| GPT-o1 | 0.5295 | 0.4628 | 0.4870 | 0.5248 | 0.4943 | 0.5322 | 0.4933 | 0.4912 | 0.5884 | 0.5474 | 0.5879 | 0.6258 | 0.5190 |
| GPT-o3 | 0.5800 | 0.5922 | 0.3868 | 0.5369 | 0.6570 | 0.7554 | 0.6595 | 0.6011 | 0.6985 | 0.7396 | 0.3952 | 0.5311 | 0.4063 |
| Gemini-2.5-Pro | 0.6047 | 0.6319 | 0.5746 | 0.5397 | 0.5934 | 0.6178 | 0.6323 | 0.6880 | 0.7734 | 0.5320 | 0.7092 | 0.4336 | 0.5305 |
| **Expert Models for Image Aesthetics Assessment** | | | | | | | | | | | | | |
| AesExpert-7b | 0.3377 | 0.3229 | 0.3146 | 0.3756 | 0.3172 | 0.4276 | 0.3582 | 0.3146 | 0.3267 | 0.3314 | 0.3803 | 0.3025 | 0.2804 |

Table 5: Evaluation on precise localization task for the bbox prediction component where the model should output coordinates of the aesthetic issues. Overall score is the average IoU of all indicators. The best and second-best results are highlighted in **bold** and underlined, respectively.

| Model | Overall Score | Layout | | | | Color | | | | Font | | Graphics | |
|---|---|---|---|---|---|---|---|---|---|---|---|---|---|
| | | balance | layering | whitespace | alignment | harmony | contrast | appeal | psycholoy | legibility | hierarchy | quality | relevance |
| **Non-reasoning Models** | | | | | | | | | | | | | |
| LLaVA-13B | 0.0559 | 0.0653 | 0.0080 | 0.0302 | 0.0172 | 0.1024 | 0.0792 | 0.0427 | 0.0303 | 0.0188 | 0.0102 | 0.1510 | 0.1160 |
| Qwen-VL-7B (Base) | 0.0514 | 0.0067 | 0.1669 | 0.0101 | 0.0259 | 0.0109 | 0.0036 | 0.0039 | 0.2306 | 0.0012 | 0.0452 | 0.0994 | 0.0063 |
| GPT-4o | 0.1712 | 0.1822 | 0.2974 | 0.1399 | 0.2552 | 0.0883 | 0.1353 | 0.0664 | 0.2186 | 0.2144 | 0.0586 | 0.2707 | 0.1270 |
| GPT-5 | **0.1993** | 0.1866 | 0.1546 | 0.2077 | 0.1613 | 0.1829 | 0.1525 | 0.1529 | 0.3348 | 0.1745 | 0.1835 | 0.2667 | 0.2338 |
| **Reasoning-augmented Models** | | | | | | | | | | | | | |
| O1 | 0.1286 | 0.0907 | 0.0767 | 0.1412 | 0.1226 | 0.1719 | 0.0204 | 0.1546 | 0.1236 | 0.1440 | 0.1441 | 0.1912 | 0.1617 |
| O3 | 0.1418 | 0.0619 | 0.1915 | 0.0552 | 0.3075 | 0.0700 | 0.0516 | 0.0760 | 0.1479 | 0.1777 | 0.2796 | 0.1861 | 0.0968 |
| Gemini-2.5-Pro | 0.0977 | 0.0518 | 0.1620 | 0.1052 | 0.0710 | 0.0760 | 0.0487 | 0.0490 | 0.2257 | 0.0963 | 0.0903 | 0.0945 | 0.1014 |
| **Expert Models for Image Aesthetics Assessment** | | | | | | | | | | | | | |
| AesExpert-7b | 0.0327 | 0.0440 | 0.0203 | 0.0093 | 0.1084 | 0.0861 | 0.0063 | 0.0049 | 0.0118 | 0.0001 | 0.0348 | 0.0601 | 0.0067 |

*Precise Localization.* As described in Section 3.3, this task consists of two components, each evaluated separately: a choice problem, where the model predicts None if no aesthetic issue exists (Table 4), and a bbox prediction problem, where the model outputs the exact bbox of the aesthetic issue (Table 5). We exclude some VLMs (e.g., Intern-VL series and small Qwen-VL models) because they failed to produce meaningful bbox predictions. For the choice problem, VLMs achieve reasonable performance, with the best model reaching an overall score of 0.6090. For the bbox prediction problem, even the best-performing model, GPT-5, scores below 0.20, highlighting the substantial difficulty of precisely localizing aesthetic issues.

*Discussions on Input Components.* When benchmarking VLMs, the input consists of three components: (1) the question, which includes a detailed explanation of the target indicator; (2) the design image; and (3) metadata in JSON format, containing layout, color, and font information. We analyze the contribution of each component to model performance using GPT-4o as a representative example. Figure 3 presents the results, where *Full Model* denotes the setting that uses all three components; *Without Images* removes the design image; *Without Explanation* omits the detailed indicator description; and *Without Metainfo* excludes the metadata. Our findings reveal three key insights. First, across all tasks, the design image is indispensable—its removal results in the largest performance drop. Second, indicator explanations have limited influence for more intuitive indicators (e.g., balance), but they play a crucial role for subjective indicators (e.g., relevance or psychology), where clearer definitions are necessary. Finally, metadata has the greatest effect on precise localization, where its absence causes a larger decline in performance compared to aesthetic judgment or region selection. We hypothesize that this is because metadata provides explicit layout information, which aids bbox prediction in localization tasks.

## 5.2 FINE-TUNING VLMS WITH AESEVAL-TRAIN

**Setups.** We construct the training set following the pipeline described in Section 4, resulting in 30k question–answer pairs. In our experiments, we use Qwen2.5-VL-7B-Instruct (Bai et al., 2025) as a representative model and adopt full-parameter finetuning on the constructed dataset. The learning rate is set to 1e-6, with a cosine scheduler and a 3% warmup ratio. For computational efficiency,

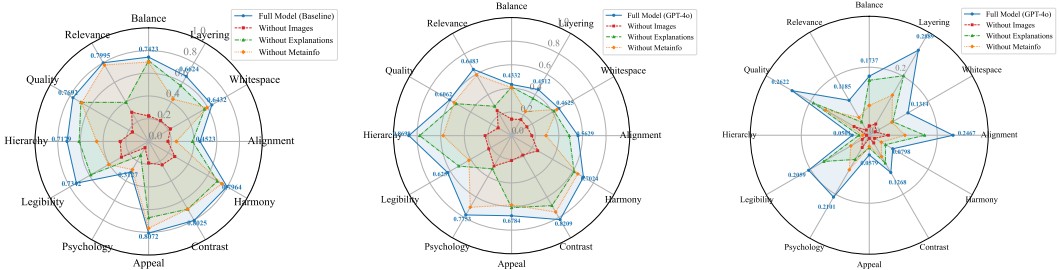

Figure 3: Results for model variants using different input components.

Table 6: Results and ablation study of fine-tuning VLMs using our constructed training set.

| Model Setting | Overall Score | Overall Gain | Overall Rank | Layout | | | | Color | | | | Font | | Graphics | |
|---|---|---|---|---|---|---|---|---|---|---|---|---|---|---|---|
| | | | | balance | layering | whitespace | alignment | harmony | contrast | appeal | psycholoy | legibility | hierarchy | quality | relevance |
| **Aesthetic Judgment (Accuracy)** | | | | | | | | | | | | | | | |
| Qwen-VL-7B (Base) | 0.6390 | - | 9 | 0.8272 | 0.4508 | 0.8076 | 0.8413 | 0.8223 | 0.9136 | 0.4009 | 0.1355 | 0.9430 | 0.3100 | 0.8183 | 0.3970 |
| + AesEval-Train | 0.6987 | + 5.97% | 4 | 0.7123 | 0.6789 | 0.7215 | 0.6868 | 0.7031 | 0.6654 | 0.7329 | 0.6577 | 0.7436 | 0.6482 | 0.7096 | 0.7244 |
| - Reasoning Path | 0.6576 | - | - | 0.6511 | 0.6589 | 0.6413 | 0.6687 | 0.6309 | 0.6791 | 0.6207 | 0.6893 | 0.6115 | 0.6985 | 0.6508 | 0.6904 |
| - Positive Samples | 0.2072 | - | - | 0.2101 | 0.1999 | 0.2202 | 0.1893 | 0.2058 | 0.2147 | 0.1955 | 0.2246 | 0.1855 | 0.2296 | 0.2004 | 0.2108 |
| **Region Selection (Accuracy)** | | | | | | | | | | | | | | | |
| Qwen-VL-7B (Base) | 0.5795 | - | 10 | 0.5128 | 0.5370 | 0.5748 | 0.5443 | 0.5822 | 0.5433 | 0.5412 | 0.6384 | 0.5974 | 0.6379 | 0.6258 | 0.6190 |
| + AesEval-Train | 0.6065 | + 2.70% | 8 | 0.5827 | 0.6108 | 0.5963 | 0.6289 | 0.5714 | 0.6236 | 0.6541 | 0.6412 | 0.6389 | 0.6487 | 0.5899 | 0.5915 |
| - Reasoning Path | 0.5795 | - | - | 0.5732 | 0.5279 | 0.5697 | 0.5322 | 0.5741 | 0.5322 | 0.5341 | 0.6283 | 0.5953 | 0.6318 | 0.6667 | 0.5885 |
| - Positive Samples | 0.5327 | - | - | 0.5089 | 0.5411 | 0.5257 | 0.5543 | 0.5013 | 0.5587 | 0.5291 | 0.5709 | 0.5212 | 0.5788 | 0.5146 | 0.4878 |
| **Precise Localization (IoU)** | | | | | | | | | | | | | | | |
| Qwen-VL-7B (Base) | 0.0514 | - | 8 | 0.0067 | 0.1669 | 0.0101 | 0.0259 | 0.0109 | 0.0036 | 0.0039 | 0.2306 | 0.0012 | 0.0452 | 0.0994 | 0.0063 |
| + AesEval-Train | 0.2231 | + 17.17% | 1 | 0.2518 | 0.1982 | 0.3103 | 0.0857 | 0.2204 | 0.2846 | 0.1552 | 0.3901 | 0.0607 | 0.2313 | 0.1152 | 0.2745 |
| - Reasoning Path | 0.0782 | - | - | 0.1523 | 0.0211 | 0.1987 | 0.0095 | 0.0750 | 0.1204 | 0.0348 | 0.0812 | 0.0159 | 0.0555 | 0.0601 | 0.1139 |
| - Positive Samples | 0.0641 | - | - | 0.1866 | 0.1546 | 0.2077 | 0.1613 | 0.1829 | 0.1525 | 0.1529 | 0.3348 | 0.1745 | 0.1835 | 0.2667 | 0.2338 |

training is performed with bfloat16 mixed precision and FlashAttention-2 (Dao et al., 2022). The vision encoder is kept frozen, while the language model parameters are tuned.

**Main Results.** Table 6 presents the results, where *Qwen-VL-7B (Base)* denotes the base model without finetuning, and *+AesEval-Train* refers to the model finetuned on our constructed training set. First, across all three tasks, finetuning with AesEval-Train yields substantial performance improvements. Moreover, on aesthetic judgment, the finetuned model surpasses even the largest Qwen-VL variant (72B parameters), and on precise localization, it outperforms GPT-5 despite the latter having far more parameters. These results demonstrate that our proposed pipeline effectively constructs training data that significantly enhances model performance.

**Ablation Studies.** We investigate the impact of different data recipes on model performance. Table 6 reports the results, where *-Reasoning Path* denotes training with plain question–answer pairs without the proposed indicator-grounded reasoning, and *-Positive Samples* denotes training only on flawed designs. We observe that *-Reasoning Path* still improves performance across all three tasks, suggesting that incorporating domain-specific knowledge of design aesthetics is beneficial. However, its performance remains notably lower than that of the full variant with reasoning paths (*+AesEval-Train*), underscoring the effectiveness of indicator-grounded reasoning. In addition, -*Positive Samples* performs worse than both *+AesEval-Train* and *-Reasoning Path*, highlighting the importance of maintaining label balance in the training set.

# 6 CONCLUSION

In this work, we introduce AesEval-Bench for design aesthetics assessment, which spans four dimensions, twelve indicators and three quantifiable tasks. Based on it, we systematically evaluate proprietary, open-source and reasoning-augmented VLMs, revealing clear gaps in design aesthetics assessment. Furthermore, we construct a training dataset to fine-tune VLMs for this domain. Experiments show that this dataset significantly improves model performance across all tasks.

**Limitations.** First, as Crello serves as the source dataset, the benchmark does not cover all types of graphic design, such as infographics or mobile UIs. Second, a fully disentangled taxonomy for indicators is not yet available. Should a more rigorously disentangled taxonomy be proposed in the future, it would be valuable to adopt it and update our indicators accordingly. Third, highly

subjective aspects of design, such as creativity, are not included. Finally, leveraging reinforcement learning to further enhance reasoning capabilities is left for future exploration.

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

**Dimension-Indicators Prompts:**

**Graphic-Quality:** Resolution is a measure of image detail. Appropriate resolution ensures the best picture quality and readability. Does element have quality issue?
**Graphic-Relevance:** Relevance refers to the direct connection between a graphic element and the meaning it conveys. Does element have relevance issue?

**Color-Harmony:** Color Harmony refers to the overall coordination, pleasure and beauty of the entire color when there are two or more colors in an image.
**Color-Contrast:** Color contrast refers to the contrasts, oppositions, and differences existing among various colors.
**Color-Appreal:** Color appeal refers to the fact that the selection and combination of colors can attract the attention of the audience.
**Color-Psychology:** Color psychology refers to the idea that color can trigger subjective psychological experiences and influence emotions, feelings, and behaviors.

**Layout-Balance:** Balance is the distribution of visual weight in design. It can be symmetrical (with equal weights on both sides) or asymmetrical (with unequal weights but still achieving visual balance).
**Layout-Layering:** Use size, color, contrast, and other visual cues to establish a hierarchical structure of design elements to guide the audience's eyes.
**Layout-Whitespace:** White space refers to the blank area around the elements in a design. Effective utilization of white space can create balance, visual hierarchy, and clarity.
**Layout-Alignment:** Alignment refers to the arrangement of design elements relative to each other or a specific axis or grid. Proper alignment creates a sense of order and organization, making the design easier to understand and navigate.

**Font-Hierarchy:** The presentation of the font has a hierarchical structure, so users can scan the text to obtain key information.
**Font-Legibility:** Legibility refers to the recognition of individual characters and the relationships between them when they are arranged side by side.

Table 7: Showcase of descriptions of each indicator.

**Task Reasoning Instructions:**

**Aesthetic Judgment :** Given the preview image and the element image and element bounding box bbox, please reason why the element is not aesthetic in the aspect of criteria. Your reason should only contain the analysis about criteria.Please think and reasoning. Once the element is mentioned, you should add its bounding box after the element(only bounding box).

**Region Selection:** Given a preview image, an image of an unsightly element and its bounding box bbox[0], and an image of an attractive element and its bounding box bbox[1], analyze why the first element is unsightly in terms of criteria based on the relationship between the elements. Your reasoning should only include an analysis of criteria. Please think and reason. Once an element is mentioned, you should add its bounding box after the element (bounding box only).

**Precise Localization:** Given a preview image, an element image, and the element's bounding box bbox, analyze why the element is unsightly when viewed from the perspective of criteria, based on their relationship. Your reasoning should only include analysis of criteria. Think and reason. Once you mention the element, you should follow it with its bounding box (only the bounding box).

Table 8: Showcase of task reasoning instructions.

## A    STATEMENT OF LLM USAGE

Large language models (LLMs) were consulted for technical guidance during implementation and debugging; following the collaborative drafting of the manuscript, we further employed LLMs to refine the prose and enhance the overall exposition.

## B    PROMPTS AND INSTRUCTIONS

First, Dimension-Indicator Prompts establish a clear set of evaluation criteria. These are organized into four core dimensions: Graphics, Color, Layout, and Font, each containing specific indicators like Font-Legibility. As shown in Table 7, every indicator is defined and paired with a guiding question to standardize the analysis.

Second, Task Reasoning Instructions (Table 8) provide operational guidance for creating the reasoning paths. They direct the analysis to focus on an element's intrinsic flaws, its relationship with other elements, or its immediate context, while critically mandating the inclusion of the element's bounding box (bbox) to ground the reasoning in precise spatial evidence.

## C    ADDITIONAL RELATED WORK

**Evaluation for VLMs.** Recent years have witnessed a surge in benchmarks Li et al. (2024b) designed to evaluate Vision-Language Models (VLMs), ranging from general-purpose assessments of perception Luo et al. (2024) and reasoning Guo et al. (2025); Zhang et al. (2025a), which encompass region-level Lin et al. (2024b; 2025); Park et al. (2025) and hierarchical understanding Kang et al. (2025); Singh et al. (2025); Zhong et al. (2025), to more specialized evaluations in domains like chemistry Li et al. (2025b) or image generation An et al. (2026). While existing benchmarks effectively evaluate general capability, they often overlook the nuanced, subjective dimensions of visual understanding. To address this, our work introduces a specialized benchmark focused on the aesthetic dimension, assessing how VLMs interpret artistic quality and visual appeal.

## D    DATA SOURCE SHOWCASE

To address the concern regarding the diversity of the Crello dataset, we provide a visual sampling of its typical data in Fig. 4. As illustrated, the dataset covers a comprehensive range of design categories and aesthetics, including textual handwritten, photography-driven and diverse art styles designs. This diversity ensures that our benchmark evaluates models on a realistic distribution of graphic design tasks, preventing bias toward any single visual style.

## E    FLAW INJECTION PIPELINE

## F    REAL WORLD DESIGN DATA SHOWCASE

To validate our model's generalization capabilities beyond the synthetic distributions of Crello, we compiled a distinct Out-Of-Distribution (OOD) test set consisting of real-world flawed designs. As shown in Fig. 5, these samples were sourced directly from intermediate design drafts and annotated by professional designers, capturing the nuanced and often complex nature of authentic.

**Mapping Real-World Flaws to Synthetic Perturbations.** To further validate the design of our data construction pipeline, we analyzed the collected real-world flawed designs (Figure 5) and mapped their defects to the operations in Algorithm 1. As observed in the "Startup Job Fair" poster, the text blends into the dark background, rendering it illegible. This specific error is simulated in our

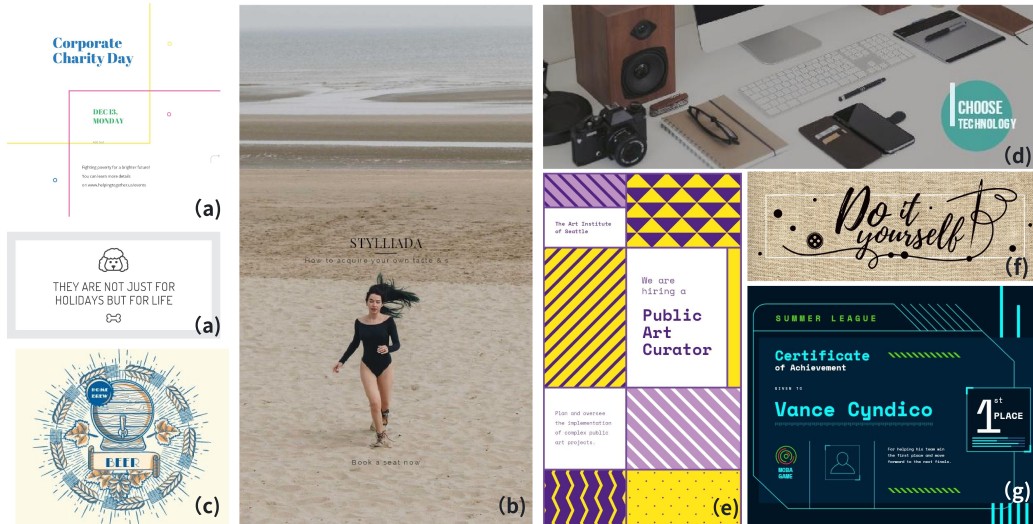

Figure 4: **Diverse design samples sourced from the Crello dataset.** The collection demonstrates a wide spectrum of visual styles and structural layouts, including (a) minimalist typography-centric designs (e.g., "Corporate Charity"), (b) photography-driven fashion editorials featuring real human subjects, (c) vintage illustrations, (d) photorealistic tech mockups, (e) geometric abstract art, (f) textured artistic typography, and (g) cyberpunk-themed certificates. This visual evidence refutes the concern of stylistic homogeneity, confirming the dataset's robust coverage across design domains.

pipeline by **Color Perturbation** module. Similarly, the "End Violence" poster exhibits severe layering issues where graphical elements obstruct critical text. This type of flaw is effectively reproduced by our **Layout Perturbation** module, which applies random coordinate shifts (Position Shift) to elements, creating unintended overlaps and layout collisions. This structural alignment confirms that our synthetic pipeline generates meaningful negatives align with the real world flaw designs.

**Note on Dataset Diversity and Benchmark Showcase.** It is important to note that for the Benchmark Showcase (Sec. I), we intentionally selected examples with isolated and obvious flaws. This selection strategy was adopted strictly for pedagogical purposes—to provide clear, unambiguous visualizations of each specific aesthetic indicator (e.g., illustrating exactly what a "Balance" violation looks like in isolation). Readers should be aware that the full AesEval-Bench and AesEval-Train datasets are significantly more diverse and challenging. They encompass a wide spectrum of difficulty, ranging from the clear, single-flaw examples shown in the showcase to complex, multi-flaw designs (similar to the real-world examples in Fig. 5) where multiple indicators (e.g., Alignment, Legibility, and Color Harmony) may be compromised simultaneously.

## G   TASK-SPECIFIC PROMPT SHOWCASES.

To ensure a rigorous evaluation of reasoning-augmented models (e.g., GPT-o1, GPT-o3), we also formulated a specific set of Optimized Prompts designed to elicit their chain-of-thought capabilities. As detailed in Tab 9, distinct from the direct queries used for standard VLMs ("Original Prompt"), these optimized prompts explicitly instruct the model to engage in a step-by-step analytical process. However, even task-specifically designed prompts could not improve the model's aesthetic understanding capability, revealing the limitations of general reasoning in this task.

## H   STATISTICS TESTS

We conducted rigorous hypothesis testing to ensure the reliability of our results reported in Table 6.

---

**Algorithm 1** Data Construction Pipeline via Synthetic Perturbation

---

**Require:** Original Design $D$, Metadata $M$ (in JSON format), Perturbation Library $\mathcal{P}$, Number of perturbed elements $n$.
**Ensure:** Perturbed Design $D'$, Updated Metadata $M'$.
 1: **Initialization:** $M' \leftarrow M$
 2: **Element Selection:** Randomly select a subset of elements $E = \{e_1, e_2, \ldots, e_n\}$ from $M'$.
 3: **for all** $e_i \in E$ **do**
 4:   **Perturbation Selection:** Randomly sample an operation $op \in \mathcal{P}$ applicable to the type of $e_i$.

 5:   **Parameter Sampling:** Sample perturbation intensity $\delta$ or target attributes.
 6:   **if** $op$ is *Layout Perturbation* **then**
 7:     $e_i.\text{pos} \leftarrow e_i.\text{pos} + \mathcal{U}(-\delta_{pos}, \delta_{pos})$ {Shift position}
 8:   **else if** $op$ is *Font Perturbation* **then**
 9:     **if** $op$ is Size Change **then**
10:       $e_i.\text{size} \leftarrow e_i.\text{size} + \mathcal{U}(-\delta_{size}, \delta_{size})$
11:     **else if** $op$ is Font Swap **then**
12:       $e_i.\text{font} \leftarrow \text{Sample}(\text{FontLibrary}) \setminus \{e_i.\text{font}\}$
13:     **end if**
14:   **else if** $op$ is *Color Perturbation* **then**
15:     **if** $op$ is Low Contrast **then**
16:       $e_i.\text{color} \leftarrow \text{SampleNear}(M.\text{background\_color}, \epsilon)$
17:     **else if** $op$ is High Contrast / Clashing **then**
18:       $e_i.\text{color} \leftarrow \text{Invert}(M.\text{dominant\_color}) + \text{Noise}$
19:     **end if**
20:   **else if** $op$ is *Graphic/Image Perturbation* **then**
21:     **if** $op$ is Replacement **then**
22:       $e_i.\text{src} \leftarrow \text{Sample}(\text{ImageLibrary})$
23:     **else if** $op$ is Resolution Reduction **then**
24:       $e_i.\text{quality} \leftarrow \text{Downsample}(e_i.\text{src}, \text{factor})$
25:     **else if** $op$ is Blur **then**
26:       $e_i.\text{effect} \leftarrow \text{GaussianBlur}(e_i.\text{src}, \sigma)$
27:     **end if**
28:   **end if**
29:   **Update:** Update element $e_i$ within metadata $M'$.
30: **end for**
31: **Rendering:** $D' \leftarrow \text{Render}(M')$ {Re-render design using updated JSON}
32: **return** $D', M'$

---

- **Aesthetic Judgment & Region Selection:** Since these are classification tasks, we applied McNemar's Test to analyze the discordance between models. The results show significant differences, with p-values of $p = 0.004$ and $p < 0.001$, respectively, confirming the effectiveness of our fine-tuning pipeline.

- **Precise Localization:** For IoU scores, we performed a Paired T-Test. Our fine-tuned model achieves a significant lead over the strongest baseline (GPT-5), with a t-statistic of $t = 4.82$ and a p-value of $p < 0.01$.

## I   BENCHMARK SHOWCASE

In this section, we provide visual examples to better illustrate the evaluation criteria for the various aesthetic dimensions within the AesEval-Benchmark. As shown in Table 10, Table 14, Table 17, Table 15, Table 13, Table 12 and Table 16, each showcase presents a side-by-side comparison of designs that exemplify positive and negative attributes for a specific criterion. These examples serve to clarify the standards used for judging aspects such as layout alignment, color harmony, graphic quality, and font legibility, offering a tangible guide to our benchmark's methodology.

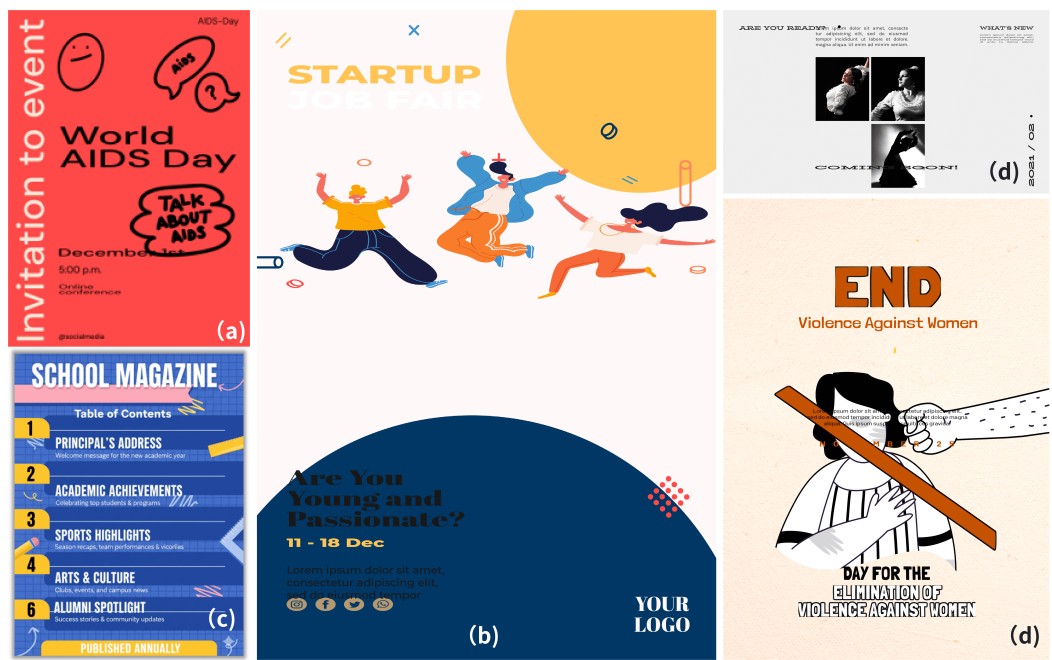

Figure 5: Representative samples from real-world flawed cases collected by professional designers. Unlike the synthetically perturbed benchmark, these designs were curated by professional designers to represent authentic aesthetic defects encountered in real-world workflows. The samples exhibit flaws such as (a) visual clutter and inconsistent orientation (e.g., "World AIDS Day"), (b) spatial imbalance and disconnected elements (e.g., "Startup Job Fair"), (c) grid-based alignment errors (e.g., "School Magazine"), and (d) typographic obstruction (e.g., "End Violence"). This dataset serves as a rigorous Out-Of-Distribution benchmark to evaluate model generalization beyond synthetic patterns.

| Task | Original Prompt | Optimized Prompt |
|---|---|---|
| **Aesthetic Judgment** | $Indicators$ + Answer it with one word 'yes' or 'no'. | Analyze the design based on $Indicators$. Evaluate the visual elements step-by-step to determine if they meet the standard. Then answer with one word 'yes' or 'no'. |
| **Region Selection** | $Indicators$ + Please only provide the index of Not aesthetic element in given bbox choices. A. [] B.[] C.[] D.[]. | Examine the candidate regions (A, B, C, D) regarding $Indicators$. Reason through the visual details to identify which specific region exhibits the flaw. Then output the index of that element. |
| **Precise Localization** | $Indicators$ + Please only provide bounding box of the Not aesthetic element... If there is no any problems, please return 'None'. | Analyze the entire design to pinpoint any element that violates $Indicators$. If a flaw is found, step-by-step determine its exact spatial coordinates. Output the bounding box in ... format, or return 'None'. |

Table 9: Comparison of Original Prompts vs. Optimized Prompts for Reasoning Models.

**Benchmark Showcase**

▷ *Layout-Alignment, Graphic-Quality*

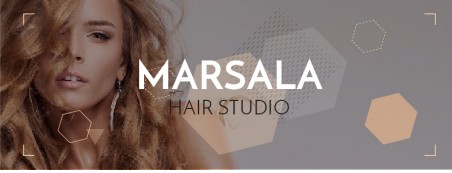 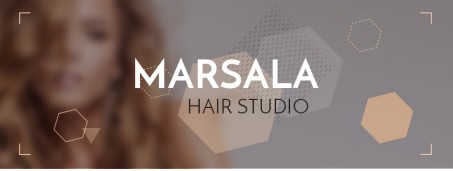

Explanation: Clear background, center-aligned text.

The background is blurry and the words in the middle are not aligned.

Table 10: Examples of Layout-Alignment and Graphic-Quality in AesEval-Benchmark.

**Benchmark Showcase**

▷ *Font-Legbility*

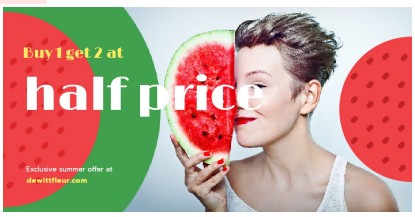 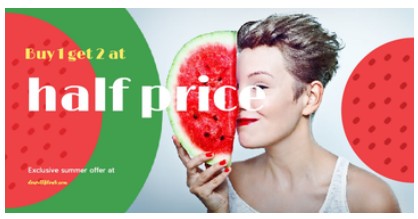

Explanation: The yellow font in the lower left corner is clearly visible.

The yellow text in the lower left corner becomes blurred and invisible.

Table 11: Examples of Font-Legbility in AesEval-Benchmark.

**Benchmark Showcase**

▷ *Graphic-Relevance*

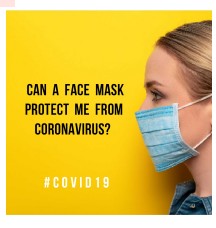 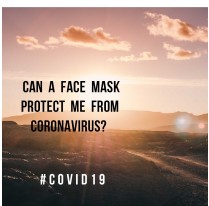

Explanation: The background is a woman wearing a mask, which is relevant.

The background is a beautiful landscape photo, which does not fit the theme.

Table 12: Examples of Graphic-Relevance in AesEval-Benchmark.

**Benchmark Showcase**

*▷ Layout-Whitespace, Layout-Layering*

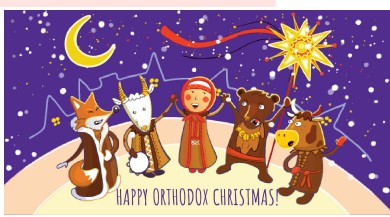 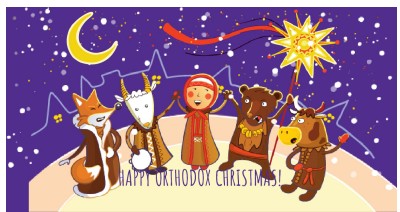

Explanation: The text is in the blank space , with no blank space or overlap.

The text has been moved to the animal's feet, resulting in white space below and stacked elements.

Table 13: Examples of Layout-Whitespace and Layout-Layering in AesEval-Benchmark.

**Benchmark Showcase**

*▷ Layout-Balance*

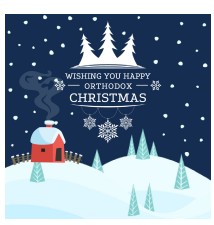 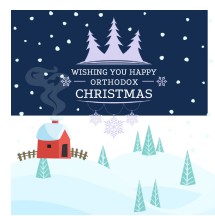

Explanation: The overall layout of the picture is balanced and coordinated.

The background of the picture is moved upwards, and the balance of the whole picture is broken.

Table 14: Examples of Layout-Balance in AesEval-Benchmark.

**Benchmark Showcase**

*▷ Layout-Hierarchy*

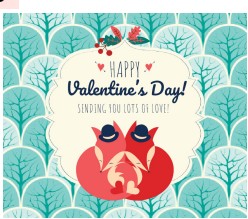 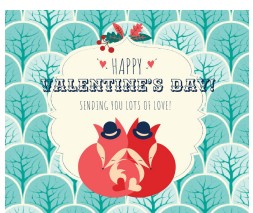

Explanation: All the fonts in the picture are consistent, which gives a sense of hierarchy.

The font in the picture is disturbed and looks like it is not on the same level as the previous font. There is no sense of hierarchy.

Table 15: Examples of Layout-Hierarchy in AesEval-Benchmark.

---

**Benchmark Showcase**

---

▷ *Graphic-Quality, Color-Contrast*

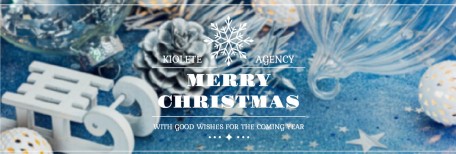 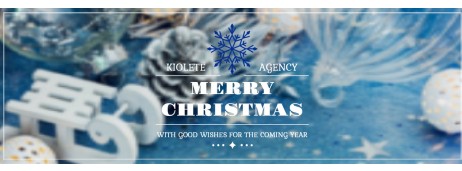

| | |
|---|---|
| Explanation: Clear background, black and white colors have good contrast. | The background is blurred and the color changes from white to black, with no contrast to the background. |

---

Table 16: Examples of Graphic-Quality and Color-Contrast in AesEval-Benchmark.

---

**Benchmark Showcase**

---

▷ *Color-Harmony, Color-Appealing, Color-Psychology*

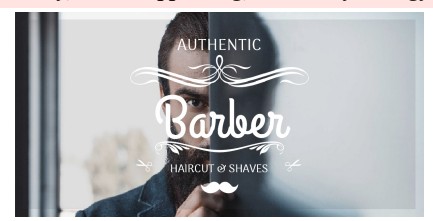 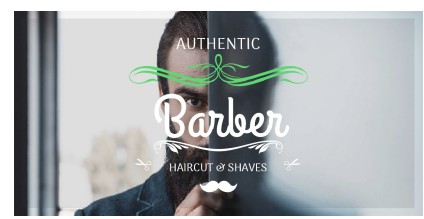

| | |
|---|---|
| Explanation: The whole picture has harmonious and beautiful colors. | The middle element turns green, which makes the whole picture look disharmonious and unappealing. Green is strange and cause bad psychological effects. |

---

Table 17: Examples of Color-Harmony, Color-Appealing and Color-Psychology in AesEval-Benchmark.

