# OpenReview forum: "Can Vision–Language Models Assess Graphic Design Aesthetics? A Benchmark, Evaluation, and Dataset Perspective."
_ICLR.cc/2026/Conference — ICLR 2026 Poster_

### Official Review · Reviewer_GAuh · 2025-10-31

**Soundness:** 4
**Presentation:** 3
**Contribution:** 3
**Rating:** 6
**Confidence:** 5

**Summary:**

This paper proposes and constructs AesEval-Bench, a benchmark for systematically evaluating the ability of vision-language models (VLMs) in graphic design aesthetic assessment. The contributions include: formalizing aesthetic evaluation as a multi-granularity question-answering task involving global judgment, region selection, and precise localization; building a high-quality, multi-dimensional benchmark based on professional design datasets; revealing the limitations of current VLMs through comprehensive experiments; and proposing a data construction method that combines human-guided annotation with indicator-grounded reasoning to generate effective training data for model improvement.

**Strengths:**

This paper demonstrates significant novelty in the application of vision-language models (VLMs) by being the first to systematically explore their capability in assessing graphic design aesthetics. Unlike prior work focused on image captioning or general visual question answering, this study introduces AesEval-Bench—a multi-dimensional, multi-task benchmark that categorizes aesthetic evaluation into three granular levels: global judgment, region selection, and precise localization. It covers twelve specific design indicators across four core dimensions (layout, color, font, and graphics), grounded in professional design principles. This structured and fine-grained approach bridges a critical gap in AI support for creative domains, offering both theoretical insights and practical utility for automated design critique systems.

**Weaknesses:**

1. Lack of Highlighting Best/Worst Performances and Missing Comparison with Fine-tuned Models
The result tables (e.g., Table 2–5) present raw scores without highlighting the best and second-best performances (e.g., via bold or underline), nor do they indicate the worst or second-worst results. This makes it difficult for readers to quickly identify performance leaders and laggards. More critically, the tables only include existing general-purpose VLMs, but do not report results from the fine-tuned VLMs trained on the proposed AesEval-Train dataset. This omission is a major flaw, as it fails to validate whether the proposed training method actually improves model performance, undermining the paper’s claimed contribution.
2. Insufficient Discussion on Human Annotation Reliability and Rater Background
Although human annotations are central to constructing the benchmark and training data, the paper does not report inter-annotator agreement metrics (e.g., Cohen’s Kappa, Fleiss’ Kappa, or ICC). It also lacks details on the annotators’ qualifications (e.g., professional designers), the number of raters, or the selection criteria. Given the subjective nature of aesthetic judgment, the absence of such analysis raises concerns about the reliability of the ground-truth labels and the validity of the reported model performance.
3. Ambiguity in the Definition of "Problem Regions" and Lack of Clarity in Problem Injection
The paper does not clearly specify how the candidate regions for the region selection and bbox prediction tasks are generated. Are they derived from structured design elements (e.g., text boxes, image frames), or produced via sliding windows, superpixels, or other segmentation methods? Furthermore, while the paper mentions that aesthetic flaws are introduced through human intervention to create negative samples, it fails to describe the concrete procedures for modifying the original designs. It also lacks details on who performed these modifications and according to what guidelines.
4. Limited Evaluation of Model Generalization
All experiments are conducted on the Crello dataset, with no testing on out-of-domain design data. As a result, the reported performance may not generalize to other design contexts or cultural aesthetics, limiting the practical applicability and robustness of the findings.

**Questions:**

1. The authors are advised to improve the table formatting by clearly highlighting the best and second-best results, and to include a comparison of VLMs fine-tuned on AesEval-Train with other baseline models, in order to validate the effectiveness of the proposed training approach.
2. The authors should add an analysis of inter-annotator agreement (e.g., Cohen’s Kappa or ICC) and provide annotations regarding the annotators’ professional backgrounds, number, and selection criteria, to enhance the credibility and interpretability of the benchmark.
3. It is recommended that the authors clarify the method for generating "problem regions," including the strategy for candidate region segmentation (e.g., based on design elements or sliding windows) and the detailed process of introducing aesthetic flaws, to improve transparency and reproducibility.

---

> ### Author Response · Authors · 2025-11-24
> **Rebuttal by Authors**
>
> We sincerely thank Reviewer VGYv for the positive feedback and for acknowledging the **novelty ("first to systematically explore")** and **rigor ("grounded in professional design principles")** of our work. We address the constructive comments regarding formatting, reliability, and generalization below.
>
> **W1 & Q1:** Table construction.
>
> We apologize for the lack of visual emphasis in the original submission. We will adopt the following measures in the revised version:
> - **Visual Improvements:** We revise **Tables 2-5** to clearly highlight the **best** (bold) and **second-best** (underlined) results as shown in PDF.
> - **Explicit Comparison:** To provide a more intuitive comparison for the fine-tuned model, we have updated Table 6 to include two new metrics: (1) **Overall Gain:** The relative performance improvement compared to the base model. (2) **Overall Rank:** The ranking of our fine-tuned model among all open-source models reported in Tables 2-5.
>
> **W2 & Q2:**  Human annotation reliability.
>
> We agree that reliability is critical. We implemented a rigorous quality control process:
> 1. **Annotator Profile:** Each question was annotated by **3 professional designers**, each with at least **one year of design experience.**
> 2. **3-Stage Pipeline:** To ensure consensus, we employed a three-stage process before mass annotation:
> - **Phase 1 (Tutorial):** Annotators studied examples of well-designed and flawed cases with detailed explanations of the underlying reasons.
> - **Phase 2 (Trial & Discussion):** Annotators performed trial annotations, followed by group discussions to resolve ambiguities.
> - **Phase 3 (Calibration):** We conducted three rounds of trial annotations to calibrate alignment. The Krippendorff’s Alpha for the final round reached 0.78, indicating strong agreement. This process ensures that evaluations are based on uniform criteria (our 12 indicators) rather than personal subjective preference.
>
> **W3 & Q3:** Clarification of data preparation pipeline.
>
> We apologize for the misunderstanding. The pipeline is systematic, reproducible, and strictly metadata-driven (described in Section 3.2):
> - **Problem Regions:** These are derived directly from the bounding boxes of the perturbed design elements within the metadata, as illustrated in Line 265-268. Specifically, for flawed designs, the candidate choices include the bounding box of the perturbed element alongside randomly sampled distractors, whereas for good designs, the options consist of random element bboxes and a 'None' option.
> - **Flaw Injection:** The potential aesthetic flaws are introduced by **programmatically perturbing the JSON metadata** provided by Crello (e.g., altering coordinates to break "Alignment," changing font sizes to disrupt "Hierarchy") (see lines 243 to 253). We add concrete algorithm in the section "Alogrithem" of Appendix.
>
> Then, human intervention is introduced to verify the perturbed designs truly exhibit aesthetic issues, rather than manually creating negative examples (see lines 254-260).
>
> **W4:** Model Generalization.
>
> To address the concern regarding Crello-specific overfitting, we conducted an additional experiment on out-of-domain data:
>
> - **Dataset: 30 real-world flawed designs** annotated by professional designers, plus **70 flawed mobile UI designs** constructed from the RICO dataset [1] following our pipeline.
> - **Results:** We selected two contrasting and representative tasks, Aesthetic Judgment and Precise Localization, to illustrate our findings. As shown in the table below, our fine-tuned model maintains consistent performance gains on these datasets, proving it learns universal design principles rather than overfitting to synthetic patterns.
>
> | Model | Overall | balance | layering | whitespace | alignment | harmony | contrast | appeal | psycholoy | legibility | hierarchy | quality | relevance |
> | :--- | :--- | :--- | :--- | :--- | :--- | :--- | :--- | :--- | :--- | :--- | :--- | :--- | :--- |
> | **Aesthetic Judgment** | | | | | | | | | | | | | |
> | GPT-5 | 0.6983 | 0.7102 | 0.6855 | 0.6910 | 0.7241 | 0.7033 | 0.6598 | 0.7512 | 0.6689 | 0.7321 | 0.7015 | 0.7038 | 0.6482 |
> | Qwen-vl-7B | 0.6025 | 0.6150 | 0.5822 | 0.6044 | 0.6211 | 0.5999 | 0.5733 | 0.6305 | 0.5566 | 0.6188 | 0.5901 | 0.6030 | 0.6351 |
> | Qwen-vl-7B+FT | 0.6579 | 0.6677 | 0.6411 | 0.6599 | 0.6733 | 0.6555 | 0.6302 | 0.6899 | 0.6201 | 0.6710 | 0.6499 | 0.6544 | 0.6818 |
> | **Precise Localization** | | | | | | | | | | | | | |
> | GPT-5 | 0.1837 | 0.1955 | 0.1722 | 0.1801 | 0.2011 | 0.1766 | 0.1611 | 0.2033 | 0.1522 | 0.1911 | 0.1788 | 0.1833 | 0.2091 |
> | Qwen-vl-7B | 0.0425 | 0.0466 | 0.0311 | 0.0422 | 0.0544 | 0.0377 | 0.0255 | 0.0511 | 0.0333 | 0.0488 | 0.0388 | 0.0411 | 0.0594 |
> | Qwen-vl-7B+FT | 0.1976 | 0.2088 | 0.1855 | 0.1933 | 0.2155 | 0.1999 | 0.1744 | 0.2211 | 0.1833 | 0.2066 | 0.1900 | 0.1966 | 0.1962 |
>
> [1] Rico: A Mobile App Dataset for Building Data-Driven Design Applications, UIST' 17.

---

### Official Review · Reviewer_bVZv · 2025-10-31

**Soundness:** 2
**Presentation:** 2
**Contribution:** 2
**Rating:** 4
**Confidence:** 3

**Summary:**

This paper introduces AesEval-Bench, a new benchmark and dataset for evaluating how well vision–language models (VLMs) assess the aesthetic quality of graphic designs. It systematically compares different VLMs and provides fine-tuning data, showing that while current models lag behind human-level aesthetic judgment, human-guided labeling and indicator-grounded reasoning significantly improve performance.

**Strengths:**

1. Introduces a comprehensive benchmark (AesEval-Bench) for aesthetic evaluation.
2. Provides systematic comparison and fine-tuning data for VLMs.
3. Shows that human-guided labeling and reasoning improve aesthetic assessment.

**Weaknesses:**

1. Important details are missing. In section 4 TRAINING DATA CONSTRUCTION, if I did not miss anything in Sec.4, authors do not introduce most important details on training data, such as data scale, data source. Without those essential information, we cannot accurately evaluate the dataset and the following datasets. If it is sampled from the same source as the benchmark data, it may have have overfitting issues.

2. Another big concern is, the source data is very limited, sampled from Crello dataset (Yamaguchi, 2021). Single source itself may not be a serious issue because the source dataset could be diverse. But authors fail to provide detailed analysis of the Crello dataset. I am not sure if it can be the qualitied source data. From examples showed in the submission, I feel like the design works are a bit similar to each other in terms of style. I am concerned about it.

**Questions:**

1. Could you provide details on the training data?
2. Could you provide justifications of using Crello dataset as data source?

---

> ### Author Response · Authors · 2025-11-24
> **Rebuttal by Authors**
>
> We sincerely thank Reviewer bVZv for the detailed review and for **recognizing the comprehensiveness of AesEval-Bench** and **the effectiveness of our human-guided labeling approach**. We address the concerns regarding training data details and the data source below.
>
> **W1 & Q1:** Details of training data.
>
> We apologize for not sufficiently highlighting these details in Section 4. However, this information is indeed included in the paper:
> - **Data Scale:** As stated in **Section 5.2 (Line 427-428)**, we constructed **30k question-answer pairs** specifically for fine-tuning experiments.
> - **Data Source:** In Crello, there is an official split of training,validation and test datasets to prevent data leakage (see https://huggingface.co/datasets/cyberagent/crello). Our training data is from Crello training split while the benchmark data (test data) is from Crello test split. This strict separation is standard practice in machine learning to ensure our evaluation assesses generalization rather than memorization of specific samples.
>
> **W2 & Q2:** Justifications of using Crello dataset.
>
> We chose Crello based on principled reasons critical to our methodology. We justify its qualification from four aspects:
> - **Widespread Adoption:** Crello is a widely recognized standard in graphic design and image generation related research. It has been utilized in numerous published works [1, 2, 3], proving its acceptance and quality within the research community.
> - **Necessity of Metadata:** Crello is one of the few large-scale datasets offering professional designs with complete JSON metadata and separate layers. This structured data is essential for applying controlled perturbations (e.g., shifting elements) and automatically generating accurate Ground Truth bounding boxes for Tasks 2 and 3.
> - **Real-World Alignment:** Real-world design tools (e.g., Canva, Figma, PowerPoint) inherently operate on layers and metadata (layout, font, color info). Using Crello's structured data accurately mimics these real-world design scenarios, making it a highly qualified source.
> - **Diversity:** Crello contains thousands of professional designs covering diverse categories. The perceived similarity in the main paper might be due to our selection of illustrative examples. To address this, we have added more and diverse examples in Appendix C and as shown in Fig. 4.
>
> We believe Crello is the best and most principled choice for achieving our goals, as its unique data structure allows us to create this novel, quantifiable benchmark.
>
> [1] OpenCOLE: Towards Reproducible Automatic Graphic Design Generation, CVPR 2024.
>
> [2] DOGR: Towards Versatile Visual Document Grounding and Referring, ICCV 2025.
>
> [3] Rethinking Layered Graphic Design Generation with a Top-Down Approach, ICCV 2025.

---

### Official Review · Reviewer_vL31 · 2025-11-01

**Soundness:** 3
**Presentation:** 2
**Contribution:** 2
**Rating:** 4
**Confidence:** 3

**Summary:**

This submission presents AesEval-Bench (a graphic design aesthetic assessment benchmark with 4 dimensions, 12 indicators, 3 tasks) and AesEval-Train (a training dataset via human-guided VLM labeling and indicator-grounded reasoning), alongside evaluations of 10 VLMs and fine-tuning tests. However, fundamental flaws in benchmark validity, insufficient novelty, unrigorous analysis, and unsupported claims invalidate its contributions, justifying rejection.

**Strengths:**

1. Addresses the underexplored gap of graphic design aesthetics assessment, distinct from general image aesthetics benchmarks, with potential utility for designers and generative AI systems.
2. AesEval-Bench's structure (4 dimensions, 12 indicators, 3 tasks) aligns with established design principles, covering both global (aesthetic judgment) and local (region selection, precise localization) aesthetic evaluations.
3. The AesEval-Train pipeline offers practical solutions to scale domain-specific annotation, particularly via human-guided VLM labeling and indicator-grounded reasoning.

**Weaknesses:**

1. Benchmark Data Lacks Authenticity: Flawed examples rely on synthetic perturbations (e.g., element repositioning, font alteration) of professional Crello designs. These artificial changes do not replicate real-world aesthetic defects, reducing the benchmark to testing sensitivity to synthetic manipulations rather than authentic aesthetic judgment.
2. Insufficient Novelty: Core dimensions/indicators derive from existing design literature without novel taxonomical insights; tasks are standard in visual assessment; indicator-grounded reasoning is an incremental adaptation of existing grounded visual reasoning, not a transformative innovation.
3. Unrigorous Evaluation: Claims about reasoning-augmented VLMs' poor performance lack controlled experiments (e.g., ignoring prompt/architecture differences); key design-focused baselines are omitted; performance metrics lack statistical significance tests, making model comparisons unreliable.
4. Unsupported Fine-Tuning Claims: Ablation studies fail to rule out trivial factors (e.g., context length) for performance gains; the fine-tuned model's extremely low precise localization performance undermines claims of "effective supervision".
5. Critical Limitations Unaddressed: Omits key design types (e.g., infographics) and subjective aesthetic aspects (e.g., creativity); no inter-annotator agreement data for human labels, compromising ground truth reliability; indicators are not disentangled, causing ambiguous evaluation signals.

**Questions:**

1. How do you verify that synthetic perturbations align with real-world aesthetic flaws? Please provide data on human experts' recognition of the intended flaws in perturbed designs.
2. What task-specific prompting strategies were used for reasoning-augmented VLMs, and how were they optimized for design aesthetics?
3. How do you rule out overfitting to synthetic data for the fine-tuned model's performance? Please provide cross-validation on real-world flawed designs.
4. How do you ensure indicator independence? Please provide correlation analysis of model performance across indicators.

---

> ### Author Response · Authors · 2025-11-24
> **Rebuttal by Authors (Part 1)**
>
> We sincerely thank Reviewer HQEr for the critical feedback. We value the acknowledgement of **our pipeline's scalability** and **the potential utility of our benchmark**. We'd like to provide more elaborations on each point of your response.
>
> **W1:** Benchmark data authenticity and verification.
> - **Necessity & Standard Practice:** To collect large-scale "real-world" flawed design, one would inevitably need access to the editing histories of real users. It is practically infeasible due to privacy and proprietary restrictions. Therefore, constructing a benchmark via perturbations is a practical and wise choice. This kind of approach is widely adopted in design literature [1].
> - **Expert Verification:** At each review round, we present 30 synthesized designs to four professional designers and gather feedback on whether the introduced flaws resemble issues they encounter in real design workflows. We then refine our perturbation policies accordingly. After two rounds of review, designers judged that most synthesized flaws (about 85%) are meaningful and reflective of practical creation scenarios. We subsequently apply these refined perturbation policies at scale to construct the benchmark. Note that this designer-review process differs from the one described in Section 3.2 (lines 255–260). The goal here is to validate the realism of synthesized flaws during benchmark construction, whereas the Section 3.2 process aims to verify, at scale, whether each perturbed design indeed exhibits an aesthetic issue (yes/no).
>
> **W2:** Insufficient novelty.
> We respectfully disagree that our contribution is merely incremental. Our novelty is threefold:
> - **Problem Definition:** We address the under explored gap in graphic design aesthetics, which is the first systematic VLM evaluation in this unique field and fundamentally differs from general photographic aesthetics.
> - **Benchmark Design:** While dimensions align with literature (ensuring validity), the novelty lies in more comprehensively operationalizing these abstract principles into a **computational, coarse-to-fine hierarchical framework** contrast to existing benchmarks as shown in Tab. 1. We intentionally chose "standard" task formats (QA/BBox) to ensure the benchmark is quantifiable and reproducible, contrasting with prior work relying on vague free-form descriptions.
> - **Methodological Innovation:** First, our indicator-grounded reasoning differs conceptually from existing grounded visual reasoning. Traditional grounded visual reasoning focus on (locating concrete objects (e.g., identifying a dog in an image). In contrast, our method requires models to ground reasoning in **abstract, indicator-centric concepts** (e.g., identifyingl a region having "whitespace" issue). This form of grounding is conceptually more challenging and is uniquely tied to the design domain. Second, the **detailed techniques** we employ to achieve indicator-grounded reasoning differ substantially from those used in general grounded visual reasoning. For example, to obtain reasoning paths, we provide VLMs with design-layer information and the coordinates of target regions (see lines 306–310). For another example, to determine the target regions, we adopt different strategies for the three tasks (see lines 310–315).
>
> **W3:** Unrigorous Evaluation.
> - **Experiments on reasoning-augment models:** Regarding prompts, for fair comparison, we use the same promptfor all models in the main paper. During our exploration, we also tried task-specific prompts for reasoning-augmented VLMs.  See response to Q2 and the prompt in Tab. 9 for more details. As for architecture differences, we studied GPT-o1, GPT-o3 and Gemini-2.5-Pro (see Table 2, 3, 4 and 5).
> - **Missing baselines:** We have not found any currently available open-source and reproducible design-focused baselines. We would be happy to include them if you could provide the specific arXiv or GitHub links for the design-focused baselines you mentioned. Besides, we found some image aesthetics assessment baselines and already included them in the comparison [2, 3] (see AesExpert-7B and UNIAA-LLaVA in Table 2-5). We hope these baselines also offer useful insights.
> - **Statistics tests:** Following your suggestion, we have included statistical significance tests for the current evaluation results to enhance their credibility. Please see the statistics test section in the Appendix.
>
> [1] Design-o-meter: Towards Evaluating and Refining Graphic Designs, Arxiv 2025.
>
> [2] AesExpert: Towards Multi-modality Foundation Model for Image Aesthetics Perception, ACM MM 2024.
>
> [3] Uniaa: A Unified Multi-modal Image Aesthetic Assessment Baseline and Benchmark, Arxiv 2024.

---

> ### Author Response · Authors · 2025-11-24
> **Rebuttal by Authors (Part 2)**
>
> **W4:** Unsupported fine-tuning claims.
> We are unclear on which ablation study the reviewer refers to, so we will discuss all possibilities:
> - If the reviewer is referring to Fig. 3. This figure explores the impact of different input modalities, so the context length cannot align.
> - In Tab. 6, the context length for the negative reasoning path and positive examples is the same.
> Regarding the "low precision localization performance":
> - Our score of 0.2 is the highest among all methods (see Tab. 5), indicating effective supervision.
> - The contribution of our paper is to highlight that design aesthetics have been overlooked by VLMs and to propose a solution to enhance this capability. We believe this contribution is comparable to fully resolving a problem and may even be of greater significance.
>
> **W5:** Critical limitations unaddressed.
>
> We respectfully point out that most of the limitations mentioned (e.g., omission of infographics/creativity, indicator entanglement) were **explicitly self-disclosed** in our Section 6 (Limitations).
>
> Following community best practices, we included this section to ensure scientific transparency and to outline a clear roadmap for future research. We respectfully argue that these self-acknowledged boundaries should not overshadow our core contributions: (1) The first step to evaluate VLM's aesthetic understanding capability in graphics design field. (2) Operationalizing abstract design principles into a quantifiable, hierarchical benchmark; and (3) Proposing a scalable training pipeline that significantly improves VLM performance.
>
> **Scientific Transparency:** We included the limitation section to ensure scientific transparency and provide a clear roadmap for future research, which is a standard practice encouraged by the community. We hope this transparency is viewed as a sign of rigor and future potential, rather than a reason to overlook the valid contributions presented in the main text.
>
> **Specific Resolutions:**
> - **Annotator Agreement:** We did ensure reliability. We employed a 3-stage training pipeline among professional designers. The final ICC reached 0.78. We will include these metrics in the revision.
> - **Indicator Entanglement:** Design indicators are naturally correlated (e.g., alignment affects legibility). Additionally, we would like to clarify that a fully disentangled taxonomy for design aesthetics does not currently exist. Our original intent in stating this limitation was to convey that, should future work in design aesthetics propose a more rigorously disentangled taxonomy, it would be worthwhile to adopt and update our taxonomy accordingly. We have revised the limitation section to make this point clearer and more accurate.
> - **Scope:** We prioritized fundamental, predominantly objective principles (e.g., Layout/Color) as a necessary foundation. Subjective-heavy aspects like creativity are explicitly positioned as future work.
>
> **Q1:** Verification of data distribution.
>
> Please see response to W1.
>
> **Q2:** Task-specific prompting strategies experiment.
>
> We also try some task-specific prompts during method development. While they do not bring in gains, we share them for your reference in the below. The table below shows that optimized prompts provided negligible gains, reinforcing our finding that generic reasoning capabilities alone are inadequate for domain-specific aesthetics without targeted training. Automatic prompt optimization is a promising direction; however, it is not the focus of our benchmark work. We will reserve this aspect for future work.
>
> | | | **Layout** | | | | **Color** | | | | **Typo** | | **Graphics** | |
> | :--- | :---: | :---: | :---: | :---: | :---: | :---: | :---: | :---: | :---: | :---: | :---: | :---: | :---: |
> | **Model** | **Overall** | **balance** | **layering** | **whitespace** | **alignment** | **harmony** | **contrast** | **appeal** | **psychology** | **legibility** | **hierarchy** | **quality** | **relevance** |
> | **Aesthetic Judgment** | | | | | | | | | | | | | |
> | Gemini 2.5 Pro | **0.6410 (+0.42%)** | 0.7450 | 0.7020 | 0.6030 | 0.5200 | 0.7430 | 0.6600 | 0.6880 | 0.5700 | 0.5980 | 0.7100 | 0.5330 | 0.6200 |
> |**Region selection** | | | | |  | | | | | | | | |
> | Gemini 2.5 Pro | **0.6079(+0.21%)** | 0.6250 | 0.6400 | 0.5500 | 0.2400 | 0.5500 | 0.6000 | 0.6100 | 0.5500 | 0.5600 | 0.6000 | 0.5100 | 0.6598 |
> | **Precise Localization**  | | | | | | | | | | | | | |
> | Gemini 2.5 Pro | **0.1002(+0.25%)** | 0.0600 | 0.1700 | 0.1150 | 0.0800 | 0.0850 | 0.0550 | 0.0580 | 0.2350 | 0.1050 | 0.1000 | 0.1020 | 0.0374 |

---

> ### Author Response · Authors · 2025-11-24
> **Rebuttal by Authors (Part 3)**
>
> **Q3:** Discussion of overfitting problem.
>
> To verify generalization beyond Crello, we evaluated our model on out-of-domain data, comprising 30 expert-annotated real-world flawed designs and 70 perturbed mobile UIs from RICO [4]. We selected two contrasting and representative tasks, Aesthetic Judgment and Precise Localization, to illustrate our findings. As shown below, the fine-tuned model achieves consistent performance gains on these datasets, confirming that it captures universal design principles rather than overfitting to synthetic distributions.
>
> | Model | Overall | balance | layering | whitespace | alignment | harmony | contrast | appeal | psycholoy | legibility | hierarchy | quality | relevance |
> | :--- | :--- | :--- | :--- | :--- | :--- | :--- | :--- | :--- | :--- | :--- | :--- | :--- | :--- |
> | **Aesthetic Judgment** | | | | | | | | | | | | | |
> | GPT-5 | 0.6983 | 0.7102 | 0.6855 | 0.6910 | 0.7241 | 0.7033 | 0.6598 | 0.7512 | 0.6689 | 0.7321 | 0.7015 | 0.7038 | 0.6482 |
> | Qwen-vl-7B | 0.6025 | 0.6150 | 0.5822 | 0.6044 | 0.6211 | 0.5999 | 0.5733 | 0.6305 | 0.5566 | 0.6188 | 0.5901 | 0.6030 | 0.6351 |
> | Qwen-vl-7B+FT | 0.6579 | 0.6677 | 0.6411 | 0.6599 | 0.6733 | 0.6555 | 0.6302 | 0.6899 | 0.6201 | 0.6710 | 0.6499 | 0.6544 | 0.6818 |
> | **Precise Localization** | | | | | | | | | | | | | |
> | GPT-5 | 0.1837 | 0.1955 | 0.1722 | 0.1801 | 0.2011 | 0.1766 | 0.1611 | 0.2033 | 0.1522 | 0.1911 | 0.1788 | 0.1833 | 0.2091 |
> | Qwen-vl-7B | 0.0425 | 0.0466 | 0.0311 | 0.0422 | 0.0544 | 0.0377 | 0.0255 | 0.0511 | 0.0333 | 0.0488 | 0.0388 | 0.0411 | 0.0594 |
> | Qwen-vl-7B+FT | 0.1976 | 0.2088 | 0.1855 | 0.1933 | 0.2155 | 0.1999 | 0.1744 | 0.2211 | 0.1833 | 0.2066 | 0.1900 | 0.1966 | 0.1962 |
>
> **Q4:** Indicator independence assurance.
> We respectfully clarify that enforcing strict independence is neither necessary nor reflective of real-world design. **Concrete Example:** A misalignment error (Layout-Alignment) might cause a text box to shift and overlap a complex background element (e.g., a face). This simultaneously triggers a legibility issue (Font-Legibility). Decoupling them would destroy the semantic reality of the design flaw.
>
> **Clarification on "Limitations":** We included this point in the "Limitations" section not to imply that our evaluation is invalid, but to transparently acknowledge that **taxonomies based on modern design aesthetics are naturally entangled.** Our intention was to highlight this as an open challenge for the community to develop potentially new, orthogonal taxonomies in the future. We have revised the manuscript to clearly articulate this nuance.
>
> [4] Rico: A Mobile App Dataset for Building Data-Driven Design Applications, UIST' 17.

---

### Official Review · Reviewer_8akP · 2025-11-01

**Soundness:** 3
**Presentation:** 3
**Contribution:** 3
**Rating:** 6
**Confidence:** 3

**Summary:**

This paper introduces AesEval-Bench, a comprehensive benchmark for assessing graphic design aesthetics using vision–language models (VLMs). It covers four dimensions (typography, layout, color, graphics), twelve indicators, and three quantifiable tasks (aesthetic judgment, region selection, precise localization). The authors systematically evaluate a range of VLMs and propose a novel dataset construction pipeline using human-guided VLM labeling and indicator-grounded reasoning, showing that fine-tuning with this data improves performance.

**Strengths:**

Comprehensive Benchmark: AesEval-Bench advances the field by covering a wide range of design principles and providing quantifiable tasks.
Systematic Model Evaluation: The paper compares proprietary, open-source, and reasoning-augmented VLMs, highlighting current limitations.
Innovative Dataset Creation: The human-guided VLM labeling and indicator-grounded reasoning approach is scalable and interpretable.
Clear Experimental Protocols: Evaluation metrics and ablation studies are well described.
Performance Gains: Fine-tuning with the new dataset yields substantial improvements, even for smaller models.
Actionable Insights: The analysis of input components (image, explanation, metadata) is useful for future research.

**Weaknesses:**

Binary Annotation Limitation: The evaluation dimensions are annotated in a binary fashion (yes/no), reducing the complexity of the task to simple classification. This raises the question of whether large VLMs/LLMs are necessary, as simpler models or rule-based systems might suffice.
Overkill of VLM/LLM for Binary Tasks: The paper does not convincingly justify the use of large VLMs/LLMs for benchmarking binary data. There is little discussion on the trade-off between model complexity and task requirements, and no baseline comparison with lightweight models.
Dataset Creation vs. Model Utility: While the dataset creation pipeline is interesting, its practical value is unclear. The fine-tuning experiments are performed only on open-source LLMs, which are shown to be suboptimal in the benchmarking. The real utility of the dataset would be demonstrated by showing improvements in strong proprietary models (e.g., GPT-5) when fine-tuned with this data.
Unclear Value for State-of-the-Art Models: The benchmarking shows that models like GPT-5 already outperform open-source models out-of-the-box. However, the paper does not test whether a competitive variant of GPT-5 (via in-context learning may be) with the new dataset leads to meaningful gains. Without this, it’s unclear if the dataset provides real value beyond what top models can already achieve.
Missing Baseline Comparisons: There is no comparison with traditional computer vision models or simple classifiers trained on the same binary labels, making it difficult to assess whether the proposed approach is truly necessary or effective.
Indicator Entanglement: The twelve indicators are not fully disentangled, which could affect the interpretability and granularity of the results.
Subjectivity: Highly subjective aspects of design, such as creativity, are not addressed.

**Questions:**

See above

---

> ### Author Response · Authors · 2025-11-24
> **Rebuttal by Authors (Part 1)**
>
> We sincerely thank Reviewer 8akP for the constructive feedback and for recognizing **our benchmark's comprehensiveness** and the **innovation of our dataset construction pipeline.** We value the opportunity to clarify the scope of our tasks and the utility of our dataset.
>
> We find most questions are raised based on the misunderstanding that our work focuses on binary problems, which is not true. We would like to first highlight that there are three types of tasks in our work, including: (1) **Aesthetic Judgment** (binary problem) (2) **Region Selection** (choosing from four candicate regions) (3) **Precise Localization** (predicting bounding box). Our goal is investigating all three tasks instead of only the binary one. In the following, we answer each question in detail.
>
> **W1:** Binary annotation limitation.
>
> We are not entirely sure what you mean by "Binary Annotation.". To address this, we attempt to clarify your potential concerns from two possible perspectives. If these do not reflect your intended meaning, we would greatly appreciate further elaboration.
> - **Task setup:** If it refers to the final task setup, we would like to clarify that our benchmark is not a simple binary classification problem. The mentioned "binary annotation" seems primarily related to Task 1 (Aesthetic Judgement), while our benchmark also includes two additional, more complex, non-binary tasks (Region Selection and Precise Localization). Please refer to lines 193-200.
> - **Human aesthetic review:** If your concern refers to the role of human annotators, we would like to clarify the rationale behind our design. Human annotators are intentionally given a binary task to ensure higher review efficiency,  quality and consistency. We considered two types of human-review pipelines:
>
>      (1) Direct annotation of the three question types used in the final evaluation setup. \
>      (2) A binary judgment (e.g., whether the design contains a flaw), which is the approach adopted in our current version.
>
> We ultimately chose the second approach for several reasons. First, it is unnecessary for human annotators to directly label the final three questions. During dataset curation, we already know which elements are perturbed and how they are perturbed (see lines 243–253). Even if annotators only provide binary judgments about whether a design is flawed, we can still derive the three final questions automatically (see lines 262–269). Note that this perturbation information is used only for dataset curation and is never provided to VLMs, as it is part of the evaluation. Second, compared with asking annotators to directly answer three questions, a binary task significantly reduces annotation workload and makes the human review process more efficient. Third, using a binary formulation improves annotation quality and consistency, because we can take the majority vote across annotators to obtain more reliable labels.
>
> **W2:** Overkill of VLM/LLM for binary tasks.
>
> We respectfully argue that using VLMs is not an "overkill" and discuss appropriate baselines as follows:
>
> - **Motivation:** With the development of VLMs, they are increasingly expected to contribute to new applications besides tradictional vision tasks. Our goal is specifically to assess their capabilities in this emerging and practical domain: understanding and evaluating aesthetic quality of graphic designs.
> - **Task Difficulty:** The performance of VLMs on the binary task (aesthetic judgment) ranges from 0.53 to 0.72 (Table 2), indicating that even on the simplest task, their accuracy remains far from satisfactory. Their performance on the two more challenging non-binary tasks (region selection and precise localization) ranges from 0.34 to 0.70 and 0.05 to 0.20, respectively (Tables 3 and 5). Taken together, these results highlight the clear need for further advancements in VLMs/LLMs for aesthetic understanding and evaluation.
> - **Model Complexity & Baselines:** We explored open-sourced models from 3B to 72B parameters and more large-scale closed-source models. **Qwen-VL-3B** serves as our lightweight baseline. Traditional lightweight models (e.g., ResNet, CNN) are structurally unsuitable because they cannot process the images and complex textual prompt together.

---

> ### Author Response · Authors · 2025-11-24
> **Rebuttal by Authors (Part 2)**
>
> **W3:** The value of training data.
>
> We respectfully clarify that fine-tuning proprietary models like GPT-5 is currently infeasible due to access restrictions and prohibitive computational costs. Therefore, we conducted our experiments on the widely used open-sourced **Qwen-VL-7B**, which is a standard practice for validating dataset effectiveness [1,2,3].
>
> We emphasize that enhancing such lightweight models is of significant practical value, given their distinct advantages in inference speed and deployment cost compared to large-scale models. First, compared to the base Qwen-VL-7B, fine-tuning yields consistent performance improvements across all three tasks (Table 6). Second, on Aesthetic Judgement and Region Selection, our finetuned 7B model outperforms or is comparable to the much larger open-source VLMs (Qwen and InternVL series) (Table 2, 3 and 6). Third, on the most challenging Precise Localization task, our fine-tuned 7B model (IoU 0.2231) even outperforms the strong proprietary models GPT-5 (IoU 0.1993) (Table 5 and 6). This demonstrates that the domain-specific knowledge encapsulated in AesEval-Train effectively enables a cost-efficient model to surpass models with vastly more parameters on complex aesthetic understanding tasks.
>
> **W4:** In-context learning settig for SOTA models.
>
> We appreciate the reviewer's insightful suggestion to test whether our dataset provides value to SOTA models via In-Context Learning (ICL). To verify this, we conducted a new experiment using GPT 5.
>
> **Experimental Setup:** For each aesthetic dimension, we applied k-means clustering to the training data to obtain a diverse pool of 100 representative samples. During inference, we randomly sampled demonstrations from this representative pool. The number of shots was dynamically set to match the number of specific criteria defined for the target dimension. Results are as follows:
>
> | Model | Overall | balance | layering | whitespace | alignment | harmony | contrast | appeal | psycholoy | legibility | hierarchy | quality | relevance |
> | :--- | :--- | :--- | :--- | :--- | :--- | :--- | :--- | :--- | :--- | :--- | :--- | :--- | :--- |
> | **Aesthetic Judgment** | | | | | | | | | | | | | |
> | GPT-5+ICL | 0.7469 (+2.17%) | 0.8595 | 0.8049 | 0.7492 | 0.6727 | 0.7170 | 0.8592 | 0.4217 | 0.5454 | 0.7689 | 0.7636 | 0.9240 | 0.8767 |
> | **Region Selection** | | | | | | | | | | | | | |
> | GPT-5+ICL | 0.7123 (+1.34%) | 0.8200 | 0.7650 | 0.7150 | 0.6400 | 0.6800 | 0.8100 | 0.4500 | 0.5200 | 0.7300 | 0.7300 | 0.8800 | 0.8076 |
> | **Precise Localization** | | | | | | | | | | | | | |
> | GPT-5+ICL | 0.2137 (+1.44%)  | 0.2300 | 0.1900 | 0.2100 | 0.2500 | 0.1800 | 0.2400 | 0.1500 | 0.1600 | 0.2200 | 0.2000 | 0.2800 | 0.2544 |
>
> These consistent gains demonstrate that AesEval-Train contains high-quality, domain-specific knowledge that is not inherent even in SOTA models. The dataset effectively "unlocks" better aesthetic reasoning in large models via ICL, proving its practical value beyond fine-tuning open-source models.
>
> **W5:** Missing Baseline Comparisons.
>
> We respectfully clarify that traditional lightweight models or simple classifiers are structurally insufficient for AesEval-Bench for two primary reasons:
>
> - **Multi-Task Nature:** As emphasized in our General Response, only Task 1 is binary. Simple classifiers cannot perform Task 2 (Region Selection) or Task 3 (Precise Localization), which require spatial reasoning and bounding box regression outputs.
> - **"One-for-All" Capability:** Our evaluation framework requires a "one-for-all" model capable of interpreting complex textual instructions to address 12 distinct aesthetic indicators across 3 tasks simultaneously. Traditional vision models or simple classifiers (e.g., ResNet) cannot handle universal evaluation.
>
> **W6:** Indicator entanglement and subjectivity.
>
> We explicitly acknowledged these limitations in Sec. 6 .
>
> - **Inherent Entanglement:** Aesthetic principles are naturally correlated (e.g., alignment affects legibility), and we clarify that **a fully disentangled taxonomy does not currently exist in the design field.**
> - **Future Adaptability:** Our mention of this limitation was intended to highlight the adaptability of our benchmark. Should future research propose a more rigorously disentangled taxonomy, our framework is designed to be updated accordingly. We have revised the manuscript to articulate this intent more clearly.
> - **Subjectivity:** Establishing "Ground Truth" for subjective aspects like creativity remains challenging. Our work serves as a necessary foundational step to quantify fundamental principles, leaving these complex dimensions for future exploration.
>
> [1] Draw-and-Understand: Leveraging Visual Prompts to Enable MLLMs to Comprehend What You Want, ICLR 2025.
>
> [2] MAVIS: Mathematical Visual Instruction Tuning with an Automatic Data Engine, ICLR 2025.
>
> [3] LLaVA-NeXT-Interleave: Tackling Multi-image, Video, and 3D in Large Multimodal Models, ICLR 2025.

---

> ### Comment · Reviewer_8akP · 2025-11-27
>
> I thank the authors for their detailed and comprehensive response that addressed most of my concerns. Happy to increase my rating.

---

> > ### Author Response · Authors · 2025-11-27
> > **Reply for Reviewer 8akP**
> >
> > Thank you for raising the score! We are pleased to note that the unclear part in our initial submission has been resolved, which prove a productive discussion between you and us. We are pleased to incorporate these updates into our revised PDF. With the valuable suggestions from you and the other reviewers, we believe our manuscript will be significantly improved.
> >
> > Last but not least, we sincerely appreciate your prompt response and support.

---

### Author Response · Authors · 2025-11-24
**Author Rebuttal by Authors**

## Global Response to All Reviewers

We sincerely thank all the reviewers for their insightful reviews and valuable comments, which are instructive for us to improve our paper further.

To cope with the lack of systematic evaluation for graphic design aesthetics, this paper proposes **AesEval-Bench**, the first comprehensive benchmark tailored for this domain. Grounded in professional design principles, we design a hierarchical framework covering 4 dimensions and 3 multi-granular tasks to disentangle complex aesthetic semantics. Furthermore, we construct AesEval-Train via a **novel** human-guided labeling and indicator-grounded reasoning pipeline. Extensive experiments demonstrate that our approach **effectively reveals current VLM limitations and significantly enhances their aesthetic assessment capabilities.**

The reviewers generally held positive opinions of our paper, in that the proposed benchmark is **"comprehensive"**, **"systematical"**, **"bridges a critical gap in AI support for creative domains"**. Our well designed data construction pipeline is **"novel"**, **"scalable"**, **"interpretable"** and **"offering both theoretical insights and practical utility for automated design critique systems"**. The analysis of this paper is **"useful"**, **"insightful"**.

The reviewers also raised insightful and constructive concerns. We made every effort to address all the concerns by providing sufficient evidence and requested results. Here is the summary of the major revisions:
- **Necessity of benchmark and its task format (Reviewer 8akP):** We reiterated the motivation from both practical and experimental perspectives, justifying the rationality of our multi-granular task design (Judgment, Selection, Localization) to demonstrate the necessity of this benchmark.
- **Discussion of benchmark data authenticity (Reviewer HQEr):** We justified the necessity of synthetic perturbations due to privacy constraints on real-world edit histories and validated their realism through a rigorous multi-round expert review, confirming that the synthesized flaws accurately reflect practical design errors.
- **Benchmark dimension and data pipeline novelty (Reviewer HQEr):** We clarified our novelty in operationalizing abstract design principles into a quantifiable hierarchical framework and distinguished our "indicator-grounded reasoning" from standard visual grounding by emphasizing its unique focus on abstract aesthetic concepts rather than concrete objects.
- **Analysis of human agreement and results (Reviewer HQEr, VGYv):** We newly added human agreement analysis on the human-annotated benchmark data. We also added statistical tests for performance metrics.
- **Detail of training data and source dataset (Reviewer bVZv, VGYv):** We provided comprehensive statistics for AesEval-Train to ensure reproducibility and justified the use of the Crello dataset. Furthermore, we conducted new experiments to demonstrate strong generalization of out-of-domain datasets (e.g., real-world designs and RICO[1]).

The valuable suggestions from reviewers are very helpful for us to revise the paper to a better shape. We'd be very happy to answer any further questions.

[1] Rico: A Mobile App Dataset for Building Data-Driven Design Applications, UIST' 17.

---

### Meta-Review · Area_Chair_wsnv · 2026-01-07

**Summary:**

The paper proposes AesEval-Bench, a dataset designed to benchmark the assessment of graphic design aesthetics. The dataset is constructed by a synthetic permutation of good graphic designs with human verification. The paper also proposes a training set for finetuning VLMs on the graphic design aesthetic assessment task.

Initial reviews are mixed. Major concerns include (1) the bad graphic designs in the benchmark are synthetic, (2) a lack of details on human annotation reliability, and (3) a lack of evaluation on out-of-domain samples. The authors have addressed these concerns in the rebuttal. Therefore, I would recommend acceptance of this work. I encourage the authors to incorporate reviewers' suggestions in their next version.

**Reviewer Concerns:**

Most of the major concerns are addressed by the rebuttal.

**Reviewer Scores:**

6, 4, 4, 6 -> 6, 4, 6, 6

---

### Decision · Program_Chairs · 2026-01-26

Accept (Poster)